# The importance of small artificial water bodies as sources of methane emissions in Queensland, Australia.

Alistair Grinham[1], Simon Albert[1], Nathaniel Deering[1], Matthew Dunbabin[2], David Bastviken[3], Bradford Sherman[4], Catherine E. Lovelock[5], Christopher D. Evans[6]

[1] School of Civil Engineering, The University of Queensland, Brisbane, 4072, Australia
[2] Queensland University of Technology, Institute for Future Environments, Brisbane, QLD, Australia
[3] Department of Thematic Studies–Water and Environmental Studies, Linköping University, Linköping, SE-58183, Sweden.
[4] CSIRO Land and Water, Canberra, 2601, Australia
[5] School of Biological Sciences, The University of Queensland, Brisbane, 4072, Australia
[6] Centre for Ecology and Hydrology, Environment Centre Wales, Bangor, LL57 2UW, UK

*Correspondence to*: Alistair Grinham (a.grinham@uq.edu.au)

**Abstract.** Emissions from flooded land represent a direct source of anthropogenic greenhouse gas emissions. Methane emissions from large, artificial water bodies have previously been considered, with numerous studies assessing emission rates and relatively simple procedures available to determine their surface area and generate upscaled emissions estimates. In contrast, the role of small artificial water bodies (ponds) is very poorly quantified, and estimation of emissions is constrained both by a lack of data on their spatial extent, and a scarcity of direct flux measurements. In this study, we quantified the total surface area of water bodies $<10^5$ m$^2$ across Queensland, Australia, and emission rates from a variety of water body types and size classes. We found that the omission of small ponds from current official land use data has led to an under-estimate of total flooded land area by 24%, of small artificial water body surface area by 57%, and of the total number of artificial water bodies by an order of magnitude. All studied ponds were significant hotspots of methane production, dominated by ebullition (bubble) emissions. Two scaling approaches were developed with one based on pond primary use (stock watering, irrigation and urban lakes) and the other using size class. Both approaches indicated that ponds in Queensland alone emit over 1.6 Mt CO$_2$-eq yr$^{-1}$, equivalent to 10% of the state's entire Land Use, Land Use Change and Forestry sector emissions. With limited data from other regions suggesting similarly large numbers of ponds, high emissions per unit area, and under-reporting of spatial extent, we conclude that small artificial water bodies may be a globally important 'missing source' of anthropogenic greenhouse gas emissions.

## 1 Introduction

Over the last 20 years greenhouse gas emissions studies from large, artificial water bodies such as water supplies or hydroelectric reservoirs have clearly demonstrated these are major emissions sources. Whilst carbon dioxide (CO$_2$), nitrous oxide (N$_2$O) and methane (CH$_4$) can all be emitted, the most recent global synthesis of artificial water body emissions demonstrated that when converted to CO$_2$ equivalents, CH$_4$ accounted for 80% of fluxes (Deemer et al., 2016). Increasingly sophisticated reviews have explored the magnitude of the artificial water body contribution to regional and global CH$_4$ budgets (St. Louis et al., 2000;Bastviken et al., 2011;Deemer et al., 2016). Much of the focus in reducing the uncertainty from this anthropogenic greenhouse gas source has focussed on the spatial and temporal variability in total emission rates and, in particular, the relative contribution of CH$_4$ bubbling

(ebullition) directly from the sediment (Bastviken et al., 2011). To enable large-scale emissions estimates from larger, artificial water bodies, relationships between eutrophication status and sediment temperature (Aben et al., 2017;Harrison et al., 2017) have been developed to predict both diffusive and ebullitive emission rates. However, in regional or global scaling of emissions it is important to examine the emission rates of all types and sizes of artificial water bodies (Panneer Selvam et al., 2014). Furthermore the surface area of small water bodies can be particularly difficult to quantify in national and global datasets due to their small size and large number (Chumchal et al., 2016). In addition, the peripheral areas of small water bodies regularly experience periods of inundation and no inundation as water levels change due to their relatively shallow nature and high water use rates. The changes in their inundation status may influence emission rates as has been observed for natural ponds (Boon et al 1997).  Given there are estimated to be 16 million artificial water bodies with a surface area less than 0.1 km$^2$ (Lehner et al., 2011), understanding the rates and variability in emissions from these flooded lands will be an important refinement to global CH$_4$ budgets.

The increasing urbanisation of society as well as the expansion of agriculture and commercial mining activities has resulted in a proliferation of small artificial water bodies in many parts of the globe (Renwick et al., 2005;Downing et al., 2006;Pekel et al., 2016). This is well illustrated by the example from the United States where artificial small water bodies increased from an estimated 20,000 in 1934 (Swingle, 1970) to over 9 million in 2005 (Renwick et al., 2005). These water bodies provide valuable services and are required to irrigate crops, provide water for farm stock, manage stormwater, offer visual amenity and recreational activities, and supply water for industrial processes (Fairchild et al., 2013). Small water bodies are often avian biodiversity hotspots, for example hosting an estimated 12 million water birds in a single catchment area in the Murray-Darling river system, Australia (Hamilton et al., 2017).

The creation of small artificial water bodies also represents a transformation of the landscape, referred to in the Intergovernmental Panel on Climate Change land use emission accounting procedures as 'Flooded Lands' (IPCC, 2006). Where the creation of small water bodies leads to new greenhouse gas (GHG) emissions, these emissions are considered anthropogenic in origin according to IPCC guidelines (IPCC, 2006), , and should therefore be included in Flooded Lands emissions inventories (Panneer Selvam et al., 2014). In addition, quantifying methane emission from ponds will improve our understanding of their role in the global carbon cycle. The potential of ponds as major organic carbon sinks has been established (Downing, 2010), although the stability and permanence of organic carbon trapped within ponds is critical to determining the magnitude of this sink. Loss pathways include active de-siltation (Verstraeten and Poesen, 2000), breaching of fully silted dams (Boardman and Foster, 2011) and methane emissions.

To date, the relatively few regional studies on small, artificial water bodies (hereafter 'ponds') have focussed on water and sediment dynamics rather than GHG emissions (Downing et al., 2008;Callow and Smettem, 2009;Verstraeten and Prosser, 2008;Habets et al., 2014). Studies of GHG emissions from ponds have been limited (Downing, 2010;Deemer et al., 2016) but are in agreement with assessments of larger water bodies where CH$_4$ is the dominant GHG relative to N$_2$O and CO$_2$ (Merbach et al., 1996;Natchimuthu et al., 2014). The only regional-scale study to date was undertaken in India by Panneer Selvam et al. (2014). In order to quantify the role of artificial ponds in the global CH$_4$ cycle, as well as their role as a source of anthropogenic emissions, it is necessary to obtain both estimates of CH$_4$ fluxes from a broader range of sites and also to estimate the surface area contributing to emissions. An important part of the value of building a dataset of CH$_4$ flux estimates from a broad

range of sites is determining factors that account for spatial and temporal variability in the flux. Surface area estimates can be problematic given the range of water types (small urban lakes to large irrigation ponds) that fall within the definition of 'ponds', their frequently high temporal variation in surface area, the sheer number of such water bodies, and their ongoing increase in number over time.

Here, we present the first regional-scale assessment of $CH_4$ emissions from ponds in the Southern Hemisphere and, following the assessment of Panneer Selvam et al., (2014) only the second regional assessment globally. The assessment was undertaken in the 1.85 million $km^2$ State of Queensland, Australia. Queensland provides an effective test case for the estimation of $CH_4$ emissions from ponds because i) it incorporates a high degree of spatial variability in land use and climate, from desert to humid tropics; and ii) the irregular rainfall patterns and wide spatial coverage of aerial imagery result in a large number of artificial ponds, which are relatively easy to quantify. $CH_4$ emissions from these ponds can be considered anthropogenic in origin, because past studies of rainforest and agricultural soils in the region have clearly shown these terrestrial landscapes were weak $CH_4$ sinks (ranging from -0.02 to -5 mg $CH_4$ $m^{-2}$ $d^{-1}$) prior to inundation (Allen et al., 2009;Scheer et al., 2011;Rowlings et al., 2012).

The principle objective of this study was to establish the GHG status of ponds in Queensland, Australia. Given the paucity of GHG data from ponds, this study has focussed on empirical assessments of $CH_4$ emissions from a range of pond types rather than detailed assessments of drivers of these emissions. Our assessment comprised four components:

1.   Quantify the area of ponds, relative to regional assessments of larger artificial water bodies;
2.   Quantify $CH_4$ emission rates for a wide spectrum of pond types;
3.   Determine variability in their surface area and emission rates;
4.   Determine the influence of inundation level on emission rates.

When integrated together, these components provide a robust regional assessment of anthropogenic $CH_4$ emissions for ponds in Queensland, Australia.

## 2 Methodology

### 2.1 Study area description

Queensland, the second largest state in Australia, covers a surface area of 1.85 million $km^2$ and has a population of 4.75 million people. Land use across the state is dominated by agriculture with over 80% of the total surface area utilised for grazing cattle or irrigated cropping (Fig. 2 a; QLUMP, 2018). The Queensland agriculture sector contributes more than AUD$13 billion per year to the state economy and includes 15 million cattle and sheep as well as 4,526 $km^2$ of land under irrigation (ABS, 2018). The climate is subtropical or tropical with mean annual temperatures ranging from 27.5 °C in the state's north to 15.8 °C in the southern interior. There are large gradients in rainfall across the state ranging from a mean annual rainfall of over 3,000 mm in the coastal north east to less than 100 mm in the arid western regions (Fig. 2 b). Rainfall has a distinct annual pattern with up to 80% falling during the summer months from November to April and is subject to decadal drought and flood cycles (Klingaman et al., 2013). The economic importance of agriculture coupled with the need to provide year round water supply for these activities and the lack of predictable rainfall has resulted in the proliferation of artificial water bodies across the state (Fig. A1). However, the number and surface area of ponds in Queensland is relatively unknown as there is no legal requirement to refer ponds to the state registry due to their small size. Under current state law

only dam walls in excess of 10 m and volumes above 750 ML are referable (DEWS, 2017) and the maximum reported volume for ponds in Queensland is three times less than the referable volume (< 250 ML) (SKM, 2012). This study has assumed ponds are less than 100,000 $m^2$ as this is recognised globally as the major area of uncertainty in surface area assessments (Lehner and Doll, 2004;Downing 2010) and has been identified as a threshold in global lake inventories (Downing et al., 2006;Verpoorter et al., 2014).

**2.2 Relative surface area of ponds across the region**

To determine the number and relative surface area of ponds across the Queensland, three State Government GIS databases of artificial water bodies were utilised. However these databases required additional processing to extract comparable pond data as there were inconsistencies in the format and nomenclature of feature types. The primary database used was the most recent official assessment of land use from March 2018 (QLUMP, 2018) and within the Primary land use classification of "Water" there is a secondary category of artificial "Reservoirs/dam" divided further into "Reservoirs, Water storage and Evaporation basin." The individual water body surface area is provided and all ponds ($<10^5 m^2$) were extracted from the database. Evaporation basins were excluded, as these are commonly used for salt extraction. These ponds were then compared against two State Government databases from a high resolution assessment of artificial water bodies across the state published in 2014 and 2015. Both databases are derived from aerial (10 to 60 cm orthophotography) and satellite (0.5 to 2.5 m resolution) imagery captured between 2010 and 2014. One database contains water bodies greater than 625 $m^2$ at full supply (Reservoirs – Queensland; http://qldspatial.information.qld.gov.au/catalogue/) and for water bodies less than 625 $m^2$ a second database was used (Water storage points - Queensland; http://qldspatial.information.qld.gov.au/catalogue/).

Water bodies larger than 625 $m^2$ contained individual polygons where water body surface area was provided and all water bodies less than $10^5$ $m^2$ were extracted from the database. The database of water bodies smaller than 625 $m^2$ contained point data providing only the location of water bodies and no information on their dimensions (Fig. A1 b and c). To estimate the surface area of these systems, 100 water bodies were randomly selected using the Subset Features tool in the Geostatistical Analyst toolbox in ArcGIS (Version 10.3, ESRI Inc., Redlands, California, USA). The surface area of selected water bodies was then quantified using high resolution aerial imagery (Nearmap; www.nearmap.com.au). Typical pixel resolution of 7 cm greatly improves edge detection of ponds as it can be very challenging to separate the water edge from riparian vegetation stands with coarser-scale data. Pond edges were mapped following the methodology of Albert et al., (2016) where imagery was georeferenced and the water edge was manually traced to create individual polygons for each pond. The mean surface area of all 100 polygons was then then assumed to approximate the surface area of all individual ponds within this database and the total surface area was calculated by multiplying this mean surface area by the total number of ponds.

To ensure only one water body was reported from each location, all databases were first screened to remove repeated detections of water bodies. All remaining water bodies were then summed together to calculate total surface area of ponds and this was compared to larger reservoirs to determine their relative surface area. To undertake regional scaling of pond emissions, individual ponds were sorted using two different size class classifications: Firstly, we categorised sites into the three smallest size classes ($10^2$ to $10^3$ $m^2$; $10^3$ to $10^4$ $m^2$; and $10^4$ to $10^5$ $m^2$) in the Global Reservoir and Dam (GRanD) assessment (Lehner et al., 2011). Secondly, we divided

sites into water bodies less than 3,500 m$^2$ (primarily stock dams) and larger water bodies (primarily irrigation dams and urban lakes), following the findings of Lowe et al., (2005) and SKM, (2012).

**2.3 CH$_4$ emissions from broad spectrum of pond types**

To quantify the range of emission rates from ponds, a monitoring program was undertaken from August to December 2017 across a wide spectrum of ponds including: farm dams (irrigation and stock watering), urban lakes, small weir systems (i.e. small dams leading to widening and slowing of river flows) and rural residential water supplies (Fig. 1). Stock dams, irrigation dams and urban lakes account for the vast majority of ponds across Queensland and ponds within each category were selected to represent the regional size class distribution (Fig. A2). The majority of sites were located in coastal catchments in south east Queensland, Australia as well as one urban lake and three stock dams in Central Queensland (Fig. 2 c).

There are a number of commonly used methods to assess methane emissions from water bodies depending on the pathway of interest. For the diffusive emission pathway, rates may be modelled using the thin boundary methods or directly measured using manual or automatic floating chambers (St. Louis et al., 2000). For ebullition pathways, rates can be directly measured using acoustic surveys or funnel traps (DelSontro et al., 2011). Thin boundary layer models cannot be used to quantify the ebullition pathway and acoustic surveys or funnel traps cannot be used effectively in ponds as the water depth is often too shallow (< 1 m). We chose to use floating chambers to capture both ebullition and diffusive fluxes. CH$_4$ emission rates were measured by deploying between 3 and 16 floating chambers per water body, covering both peripheral and central zones (Fig. A3). Chamber design followed the recommendations of Bastviken et al., (2015), as these lightweight chambers (diameter 40 cm, 12 L headspace volume and 0.7 kg total weight) were ideally suited to deployment in ponds where both site access and on-water deployments can be challenging (Fig. A4). The floating chambers used were designed to yield negligible bias on the gas exchange and compare well with non-invasive approaches (Cole et al., 2010;Gålfalk et al., 2013;Lorke et al., 2015).

Where possible 24 hour measurements were undertaken, however in three water bodies this was not possible, (Table A1) and here measurements lasted between 6 and 8 hours. The 24 hour deployment time was chosen to increase the likelihood to capture ebullition, which is episodic in nature, and to incorporate diel variability in diffusive emissions which can be up to a 2-fold bias (Bastviken et al., 2004;Bastviken et al., 2010;Natchimuthu et al., 2014). The use of long term deployments may underestimate diffusive fluxes, which decrease as the chamber headspace approaches equilibrium with the water. However, in contrast to CO$_2$, CH$_4$ has a long equilibration time and it has been shown that a 24 hour deployment of these types of flux chambers on lakes underestimate diffusive fluxes by less than 10% (Bastviken et al., 2010). An initial gas sample was collected at chamber deployment and a final chamber headspace gas sample after 24 hours following the Exetainer method described in Sturm et al., (2015). CH$_4$ emission rates were calculated from the change in headspace concentration over time and normalised to areal units (Grinham et al., 2011).

**2.4 Variability in surface area and emission rate**

**2.4.1 Spatial and seasonal variability across a single water body**

To gain insight into the spatial and temporal uncertainty in pond emissions we compared variability in seasonal emissions from a single site to emissions from an intensive spatial survey of multiple sites across the pond (Fig.

4). Seasonal variability in emission rates was measured at an urban lake (St Lucia 1) where monthly monitoring at a single site was undertaken across an annual cycle (Jan to Dec 2017). This pond was selected as water level remains relatively constant throughout the year and sampling would not impacted by changes in inundation status. Emissions were monitored following the same methodology as described in the preceding section, and 4 or 5 floating chambers were deployed for each sampling event. Emission rates from this seasonal study were then compared to an intensive spatial survey of the same pond (Dec 2017), where 16 chambers were deployed simultaneously for a 24 hour incubation. To better understand spatial patterns in emissions within this pond the water depth and proximity to inflow points were mapped. The bathymetric survey was conducted using a logging GPS depth sounder (Lowrance HDS7 depth sounder, Navico, Tulsa, Oklahoma, USA). Georeferenced water depth points were imported into ArcGIS and interpolated across the whole water body using the inverse distance weighting function.

**2.4.2 Variability in water surface area across all monitored ponds**

The variability in surface area of each of the 22 ponds monitored in the emissions surveys was analysed using high resolution historical imagery across all monitored water bodies. A time-series of high resolution aerial imagery over a 9 year period from 2009 to 2017 was screened for image quality and appropriate images were selected. The time series data are not consistent across the whole state, the number of discrete images for individual water bodies varied from 3 to 16. Images of individual ponds were georeferenced to a common permanent feature across all images and then the outer water edge was mapped and surface area calculated following Albert et al., (2016). The time series of surface area for individual water bodies was compared to their corresponding surface area at full supply level ($A_{FSL}$) and expressed as a percentage then grouped into three size classes based on the GRanD classification. This time period also captured the range of rainfall variability across the state with 2010 being the wettest year on record whilst 2013 to 2015 were consecutive drought years (Average rainfall; https://data.qld.gov.au/).

**2.5 Effect of inundation status on pond emissions**

The effect of inundation status on emission rates was tested on a stock dam (Gatton 4) where measurements were undertaken on peripheral areas during periods of inundation and no inundation. This pond was selected as stock dams generally experience accelerated rates of water level change due to their relatively small size compared to other pond types (Fig. A2). In addition, the construction of this pond is typical for stock dams (a shallow pit is dug out and the soil used to construct the wall and spillway) and the surface area (1,893 m$^2$) closely matched the median for all farm dams (1,586 m$^2$; Fig. A2). Emission measurements for the inundated period followed the methodology outlined above for the water body emissions survey. Three weeks later water levels within the ponds had dropped and emission measurements were repeated at the same sites which were now exposed. For these emission measurements five chambers (90 mm diameter, 150 mm length) were carefully inserted 50 mm into the ground and care was taken to minimise disturbance to the soil surface. The headspace of each chamber was flushed with ambient air to remove headspace contamination due to chamber insertion, then the sampling port of each chamber was sealed. After the deployment period, a gas headspace sample was collected and $CH_4$ concentration analysed.

## 2.6 Statistical analyses and regional scaling of emissions

Emissions rates and surface area data were analysed using a series of one-way analyses of variance (ANOVAs) with the software program, Statistica V13 (Dell Inc., 2016). Analysis of emissions rates collected during the monthly monitoring study and the inundation study used sampling month or inundation status as the categorical predictors and chamber emission rates as the continuous variable. Emission rates from individual water bodies collected during the broad survey were first pooled into four primary use categories (irrigation, stock, urban and weirs) or three different GRanD size classes and these categories were used as the categorical predictors. The primary use of each pond was provided by pond owners or managers, in the case of urban lakes that had both aesthetic and stormwater functions these were classified as urban (Table A1). 22 ponds were included in this survey with four irrigation ponds, nine stock watering ponds, seven urban ponds and two weirs. Changes in water surface area (as a percentage of $A_{FSL}$) from individual water bodies were pooled into three GRanD size classes and these categories used as the categorical predictors. Where necessary, continuous variable data were log transformed to ensure normality of distribution and homogeneity of variance (Levene's test) with post hoc tests performed using Fisher's LSD (least significant difference) test (Zar, 1984). Tests for normality were conducted using Shapiro-Wilks tests as recommended by Ruxton et al., (2015). The non-parametric Kruskal–Wallis (KW) test was used for continuous data which failed to satisfy the assumptions of normality and homogeneity of variance even after transformation. Statistical results were reported as follows: Test applied (Fisher's LSD or Kruskal-Wallis test), the test statistic (F or H) value and associated degrees of freedom with p-value.

Emissions were scaled to water body size classes following two different approaches. Firstly, emissions were grouped according to their respective GRanD size class. These match the size class of water bodies used in the emissions monitoring of this study, and the GRanD database was used in the most recent global synthesis of greenhouse gas emissions from reservoirs (Deemer et al., 2016). Secondly, water bodies less than 3,500 $m^2$ in area were assumed to be primarily stock dams and larger water bodies primarily irrigation dams (Lowe et al., 2005). To extrapolate pond emission rates to regional scales, an appropriate measure of centrality should be used. Three common measures, arithmetic mean, geometric mean and median values, were calculated for each water body category and size class. To assess the most suitable measure of centrality for water body emissions, normal probability plots of raw and log transformed emissions data were generated and tested using the Shapiro-Wilks test (Fig. A5). The emissions data from all replicate measurements fitted a log-normal (p = 0.081) but not a normal distribution (p = 0.0000) and, therefore, the geometric mean would provide the most appropriate measure of centrality for this data (Ott, 1994;Limpert et al., 2001). Fluxes were scaled to annual rates using the cumulative surface area of water bodies and the respective emissions rate for each size class using the geometric means. The variability in geometric mean was given by the exponential of the 95% confidence interval range of log transformed data. Emissions for water bodies less than 3,500 $m^2$ were scaled using stock dam rates and larger water bodies (3,500 $m^2$ to $10^5$ $m^2$) using rates obtained from irrigation dams and urban lakes. Total fluxes from respective size classes were then combined to provide regional estimates. Annual fluxes of $CH_4$ were converted to $CO_2$ equivalents assuming a one hundred year global warming potential of 34 (IPCC, 2013).

# 3 Results

## 3.1 Relative surface area of ponds

The state wide land use assessment identified 13,046 ponds across Queensland, occupying a total surface area of approximately 467 km$^2$ (Fig. 2 c). However, with the inclusion of the additional Reservoir and Water Storage Point datasets the number of ponds increased over 20 times to a total of 293,346, and the surface area more than doubled to 1,087 km$^2$. The official land use assessment of Queensland underestimates the surface area of ponds by 57%, and the total number of water bodies by more than an order of magnitude. The revised total surface area of all artificial water bodies across Queensland increased by 24% to just over 3,248 km$^2$ (Table A2).

Ponds were widely distributed across the state, but over 78% of ponds were located on grazing land, suggesting that stock dams represent the primary water body type (Fig. 2 a). Over two thirds of ponds were confined to regions of the state were rainfall isohyets were above 600 mm (Fig. 2 b) and heavily concentrated in cropping and residential areas in the central and south eastern parts of the state (Fig. 2 c). These findings highlight the importance of striving to incorporate all artificial water bodies into flooded land emission assessments; omitting water bodies below a size threshold can lead to a dramatic under-estimation of the total number of water bodies present, and a considerable underestimate of the available surface area for CH$_4$ emissions.

## 3.2 CH$_4$ emissions from ponds

All 22 water bodies monitored in this study were shown to be emitters of CH$_4$, and emission rates ranged from a minimum of 1 mg m$^{-2}$ d$^{-1}$ to a maximum of 5,425 mg m$^{-2}$ d$^{-1}$ (Table A1). Only one water body (Mt Larcom 3) had a maximum rate below the reported upper range (50 mg CH$_4$ m$^{-2}$ d$^{-1}$) for diffusive fluxes found in larger water bodies in this region (Grinham et al., 2011;Musenze et al., 2014). Mean flux rates of only four individual water bodies were below 50 mg m$^{-2}$ d$^{-1}$ (Table A1) suggesting ebullition to be the dominant emission pathway in these systems.

Grouping ponds according to their primary use resulted in no significant differences in emissions rates between irrigation dams, stock dams and urban lakes, however, weirs were significantly higher  ($F_{(3,121)} = 6.43$, p < 0.001) than all other categories (Fig. 3 a). Mean emission rates were however higher in stock water bodies (168 mg m$^{-2}$ d$^{-1}$) compared with irrigation and urban bodies (84 and 129 mg m$^{-2}$ d$^{-1}$, respectively). Weir water bodies had mean emission rates of 730 mg m$^{-2}$ d$^{-1}$, more than four times higher those of any other category (Fig. 3 a). Grouping ponds according to their GRanD size classes resulted in significantly higher emissions rates (KW-H$_{(2,121)}$ = 7.354, p < 0.05) from ponds in 10$^2$ to 10$^3$ m$^2$ size class compared to 10$^4$ to 10$^5$ m$^2$ (Fig. 3 b). Overall, mean emissions decreased with increasing size class. Note that all weir sites fell into the smallest size category.

## 3.3 Spatial and temporal variability in surface area and emission rate

### 3.3.1 Spatial and temporal variability within a single pond

Observed emissions rates from the intensive spatial study, carried out in December 2017, ranged over two orders of magnitude from under 40 mg m$^{-2}$ d$^{-1}$ to over 3,500 mg m$^{-2}$ d$^{-1}$ (Fig. 4). Emissions were highest in the shallow southwest sector of the pond, adjacent a large stormwater inflow point, as well as along the western boundary where numerous overhanging riparian trees are located along with a second stormwater inflow point (Fig. 4).

Monthly emissions were moderately variable across the annual cycle and mean rates ranged from 176 to 332 mg $m^{-2}$ $d^{-1}$. No significant difference in emissions rates (KW-$H_{(11,50)}$ = 3.56, p = 0.98) were observed between sampling events (Fig. 5). Mean rates observed during the monthly monitoring were similar to chamber rates from the intensive spatial study (274 mg $m^{-2}$ $d^{-1}$).

### 3.3.2 Variability in water surface area across all monitored ponds

Variability in water surface area is strongly related to water body size class (Fig. 6). Mean surface area within the smallest size class was only 64% of $A_{FSL}$, this increased to 84% in the intermediate size class and to 94% in the largest size class (Fig. 6). Smaller ponds had a significantly lower surface area relative to $A_{FSL}$ (KW-$H_{(2,231)}$ = 50.523, p < 0.001) compared to larger size classes and were more variable (Fig. 6). Regional emissions estimates therefore need to correct for the differences in water body surface area relative to predicted $A_{FSL}$, particularly, in the smaller size classes.

### 3.4 Effect of inundation on stock dam emissions

The water surface area of a single stock dam ranged from 395 to 2,808 $m^2$ over a 40 month period (Fig. 7 a) with an outer band of 580 $m^2$ undergoing frequent inundation cycles (May 2016 to Dec 2017 - Fig. 7 a). Emissions rates from peripheral areas during an inundated period were significantly higher (more than one order of magnitude) compared with emissions when not inundated (KW-$H_{(1,10)}$ = 6.818, p < 0.001; Fig. 7 b). In contrast emissions from central areas were over 100 mg $m^{-2}$ $d^{-1}$, more than double the peripheral area emission rates (Table A1). This modifier of rates will primarily impact emissions from smaller size classes which have greater variability in water surface area (Fig. 6). An additional implication is in the importance of designing monitoring studies where emissions rates are quantified from both peripheral and central areas for each system. Rates monitored only in peripheral areas will likely bias towards lower emissions, particularly if these have undergone recent inundation.

### 4 Discussion

### 4.1 Relative importance of pond emissions to regional flooded land inventories

The findings of this study demonstrate ponds are an underreported and important $CH_4$ emission source in Queensland, and likely also globally. These findings highlight the importance of striving to incorporate all artificial water bodies into flooded land emission assessments; omitting water bodies below a size threshold can lead to a substantial under-estimation of the total number of water bodies present, and a considerable underestimate of the available surface area for $CH_4$ emissions. Mean annual $CH_4$ fluxes from ponds for the State of Queensland ranged between 1.7 and 1.9 million t $CO_2$ eq (Table 1) depending on the scaling approach. Given ponds represent 33.5% of the total flooded lands surface area in Queensland and emission rates are equivalent to larger water bodies in the region (Musenze et al., 2014;Sturm et al., 2014), ponds represent one-third of total emissions from flooded lands in Queensland. Remarkably, mean total emissions from ponds represent approximately 10% of Queensland's land use, land use change and forestry sector (NGERS, 2015) emissions using either scaling approach.

Future regional and global emissions estimates would be greatly improved with the inclusion of ponds, as their proliferation has been noted in five continents. In the continental United States ponds have been shown to cover 20% of total artificial water body surface area (Smith et al., 2002); in South Africa there are an estimated 500,000 ponds (Mantel et al., 2010); in Czechoslovakia ponds make up over 30% of total artificial water bodies surface area (Vacek, 1983); and in India ponds are estimated to comprise 6,238 km$^2$, or over 25% of India's artificial water body surface area (Panneer Selvam et al., 2014).

## 4.2 Pond emission pathways

Emissions rates from ponds observed in this study are consistent with ebullition being the dominant pathway. Diffusive emissions from studies of three larger water bodies in the region found the upper limit for diffusive emission was 50 mg m$^{-2}$ d$^{-1}$ (Grinham et al., 2011;Musenze et al., 2014) and only five ponds had emission rates below this level. Ebullition was observed at all ponds with maximum rates all in excess of 50 mg m$^{-2}$ d$^{-1}$ with the exception of only one stock dam (Mt Larcom 3) where the maximum rate was 19 mg m$^{-2}$ d$^{-1}$. This is a consistent finding with larger water bodies in the region where ebullition has been shown to dominate total emissions (Grinham et al., 2011;Sturm et al., 2014). The relatively higher emissions from smaller pond size classes is consistent with previous observations of increased ebullition activity in shallow zones, particularly water depths less than 5 m (Keller and Stallard, 1994;Joyce and Jewell, 2003;Sturm et al., 2014). Virtually all ponds within the smaller size classes would be less than 5 m deep. In addition, ponds trap large quantities of sediment and organic material (Neil and Mazari, 1993;Verstraeten and Prosser, 2008) and these deposition zones have been identified as methane ebullition hotspots in larger water bodies (Sobek et al., 2012;Maeck et al., 2013). The pattern in emissions from the intensive spatial study in an urban lake, where shallow areas adjacent stormwater inflows were shown to be ebullition hotspots, have also been observed in larger water bodies were ebullition activity was highest adjacent to catchment inflows (DelSontro et al., 2011;Grinham et al., 2017;de Mello et al., 2017). The emissions from small weirs were clearly dominated by ebullition, which is consistent with emissions from three larger weirs where rates ranged from 1,000 to over 6,000 mg m$^{-2}$ d$^{-1}$ (Bednařík et al., 2017). Weirs intercept the primary streamflow pathways and will likely cause large quantities of catchment derived organic matter to deposit within the weir body which, coupled to the shallow nature, results in high rates of ebullition. Overall, the rates observed for all categories, except irrigation dams, were in the upper range of reservoir areal flux rates reported in global reviews (St. Louis et al., 2000;Bastviken et al., 2011;Deemer et al., 2016), reflecting the dominance of the ebullition pathway in ponds. An additional consideration for future studies of ebullition patterns in ponds stems from recent studies of reservoirs which found significant changes in ebullition intensity and ebullition distribution as water levels decrease (Beaulieu et al., 2018;Hilgert et al., 2019). Under decreasing water levels, deeper zones of ponds may begin bubbling or increase the intensity of bubbling, this could potentially offset the reduction in surface available for emissions and total emissions would remain relatively constant.

## 4.3 Challenges in scaling emissions

Efforts to develop flooded land emission inventories rely heavily on the emission rate used to scale the surface area of water bodies' within selected categories. Given the high variability in emission rates within and between individual ponds and relatively low replication, it is critical to select an appropriate measure of centrality (arithmetic mean, geometric mean or median) in order to scale regionally and globally (Downing, 2010). For rice

paddies, septic tanks, peatlands and natural waters (Aselmann and Crutzen, 1989;Dise et al., 1993;Diaz-Valbuena et al., 2011;Bridgham et al., 2006) the geometric mean has been applied.  Likewise, in this study the log normal distribution of emissions data indicated the geometric mean as the most appropriate measure and the total emission rates using this measure fell within the reported range from larger artificial water bodies in the region (Grinham et al., 2011;Sturm et al., 2014). However, the geometric mean for all water body categories and size classes were less than half of their respective arithmetic mean values (Fig. A6). For irrigation, stock and urban water bodies, geometric mean values were actually outside of 95% confidence interval limit for the arithmetic mean (Fig. A6 a and b). Geometric mean and median values were similar across all water body categories and size classes and these measures, therefore, represent conservative emissions rates from ponds. This raises an important issue with scaling ebullition dominated water bodies as there is always going to be a high likelihood of detecting a small number of very high rates which will invariably give rise to log normal data distributions. Future studies will focus on determining whether the conservative estimates generated through the use of geometric means approximate the true emissions from ponds.

**5 Future research**

Continued efforts to quantify regional pond abundance, particularly smaller size classes, should be a research priority as this will greatly improve the surface area estimate of flooded lands used for upscaling greenhouse gas emissions as well as their role in the global carbon cycle. The increased coverage, availability and resolution of satellite imagery as well as more sophisticated methods to identify water bodies (Verpoorter et al., 2014) will support these efforts. However, it is critical to continually update regional assessments as the annual increase in farm ponds has been estimated to be as high as 60% in some parts of the globe (Downing and Duarte, 2009). Regional assessments should also correct for differences in pond surface area, particularly in the smaller size classes, as this study has demonstrated actual surface area can be significantly smaller than the surface area at full supply level ($A_{FSL}$). An additional consideration is to ensure pond emission studies from different regions include all relevant ponds types. For example, the use of ponds to increase groundwater recharge is widespread across South East Asia (Giordano, 2009) and these would need to be included in regional inventories.

Increasing both the number and type of pond within each size class in emissions monitoring studies should be a research priority. This will allow increased confidence in the selection of an appropriate measure of centrality as well as reducing uncertainty in the expected range of emission rates within each pond category. When designing a monitoring study it is important to ensure emissions rates are quantified from both peripheral and central areas for each pond. This study demonstrated that measurements taken only in peripheral areas will likely bias towards lower emissions particularly in ponds that experience rapid changes in water level and, therefore, inundation status of peripheral areas. However, this was limited to a single stock dam and additional pond types and size classes must be examined before more confident generalisations can be made.

The high spatial variability in emission rates within ponds noted from this study, highlights the importance of ensuring chambers cover the widest possible spatial scale during a measurement campaign. This will increase the likelihood of detecting ebullition zones which are likely the dominant emission pathway. However, this finding was from a single urban lake and additional long term temporal studies along with high resolution spatial surveys of different pond types and size classes are required to identify the drivers of pond emission pathways. Research

into both pond surface area and $CH_4$ emission rates will allow greater understanding of their importance to flooded land emission inventories at both regional and global scales.

**Data Availability**

The data that support the findings of this study are available from the corresponding author upon request.

**Author contribution**

AG conceived, designed and conducted the study and co-wrote the manuscript; CE, CL, DB and BS conceived, designed the study and contributed to the manuscript; SA, ND and MD conducted the study and contributed to the manuscript.

**Competing interests**

The authors declare no competing financial interests.

**Acknowledgements**

We are grateful to the reviewers for their helpful comments and suggestions. We gratefully acknowledge the following for providing access to ponds: R and L Prange; G and M Gale; M Bauer; S Green; T Connolly. In addition, we are grateful for the background information regarding the primary use of the ponds. We gratefully
acknowledge M Fluggen for laboratory and logistical support.

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

**Tables**

**Table 1. Summary of Queensland small water bodies classified using two different relative size classifications. The number of water bodies, corrected surface area of size class and total mean annual emissions. Approach 1: emissions for water bodies less than 3,500 m$^2$ were assumed to be stock dams and larger water bodies irrigation dams (Fig. 3 a), Approach 2: emissions for GRanD size classes were taken from Figure 3 b, however, weir emissions were omitted as these are not relevant at the regional scale.**

| Approach 1 | | | | | |
|---|---|---|---|---|---|
| Water body size (m$^2$) | Number | Surface area (km$^2$) | Total emissions (t CO$_2$ eq yr$^{-1}$) | | |
| | | | *Mean* | *Lower limit* | *Upper limit* |
| < 3,500 | 227,397 | 243 | 507,633 | 278,205 | 926,267 |
| 3,500 to 10$^5$ | 65,949 | 844 | 1,158,069 | 782,244 | 1,714,458 |
| **Total** | **293,346** | **1,087** | **1,665,702** | **1,060,448** | **2,640,725** |
| Approach 2 | | | | | |
| Water body size (m$^2$) | Number | Surface area (km$^2$) | Total emissions (t CO$_2$ eq yr$^{-1}$) | | |
| | | | *Mean* | *Lower limit* | *Upper limit* |
| 10$^2$ to 10$^3$ | 108,526 | 50 | 97,302 | 35,436 | 267,177 |
| 10$^3$ to 10$^4$ | 163,803 | 400 | 868,201 | 513,740 | 1,467,225 |
| 10$^4$ to 10$^5$ | 21,017 | 637 | 759,247 | 462,561 | 1,246,228 |
| **Total** | | | **1,724,749** | **1,011,736** | **2,980,629** |

**Figures**

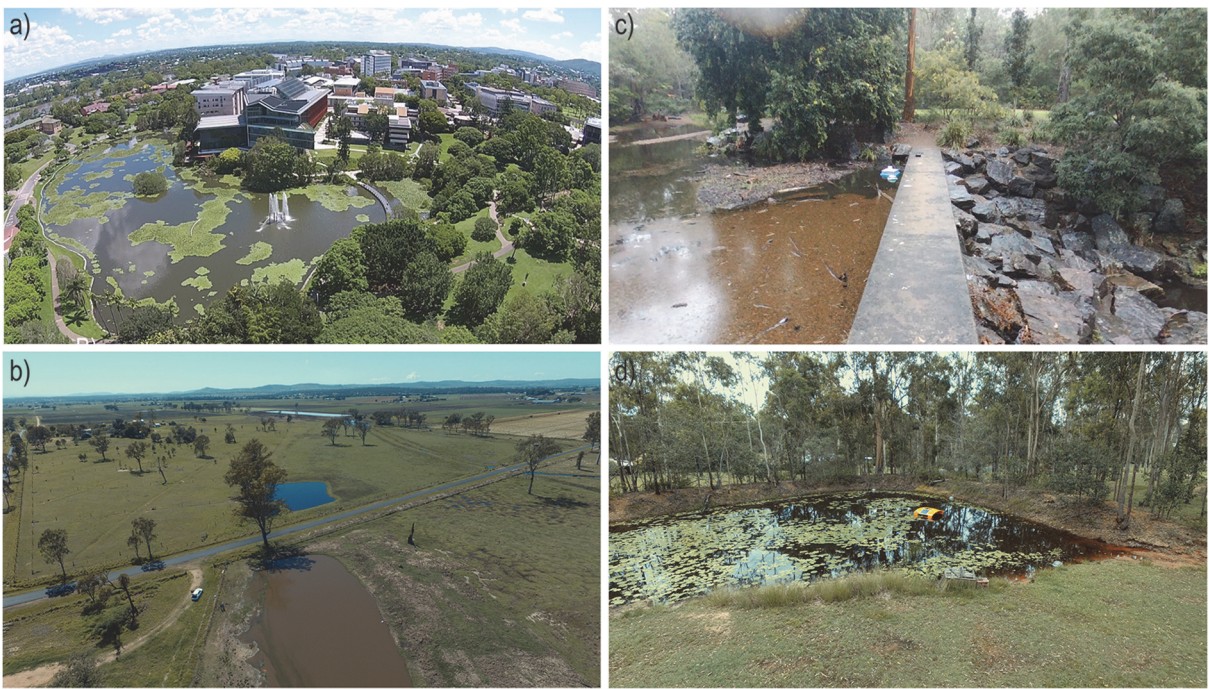

**Figure 1. Oblique drone images showing examples of ponds where CH$_4$ emissions were monitored during this study: a) urban lake (St Lucia 1); b) stock dams in foreground (including Gatton 4), irrigation dam in background; c) small weir showing high organic loading upstream of wall (Mt Cootha); d) rural residential dam (Greenbank).**

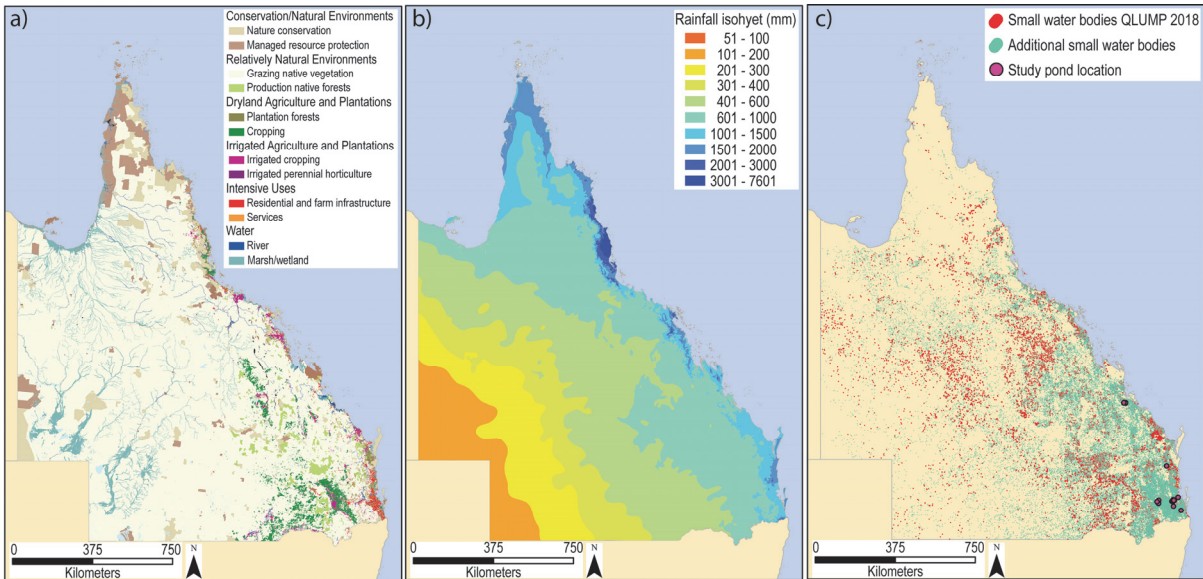

**Figure 2. a) 2018 state wide assessment showing the relative surface area occupied by secondary land use categories (QLUMP, 2018). Note the legend shows the two largest land uses within each category. b) Mean annual rainfall isohyets across Queensland from 30 period of 1961 to 1990 (http://www.bom.gov.au accessed March 2018). c) Location of study ponds and ponds identified from the land use assessment (QLUMP 2018) and two additional state wide databases (see text).**

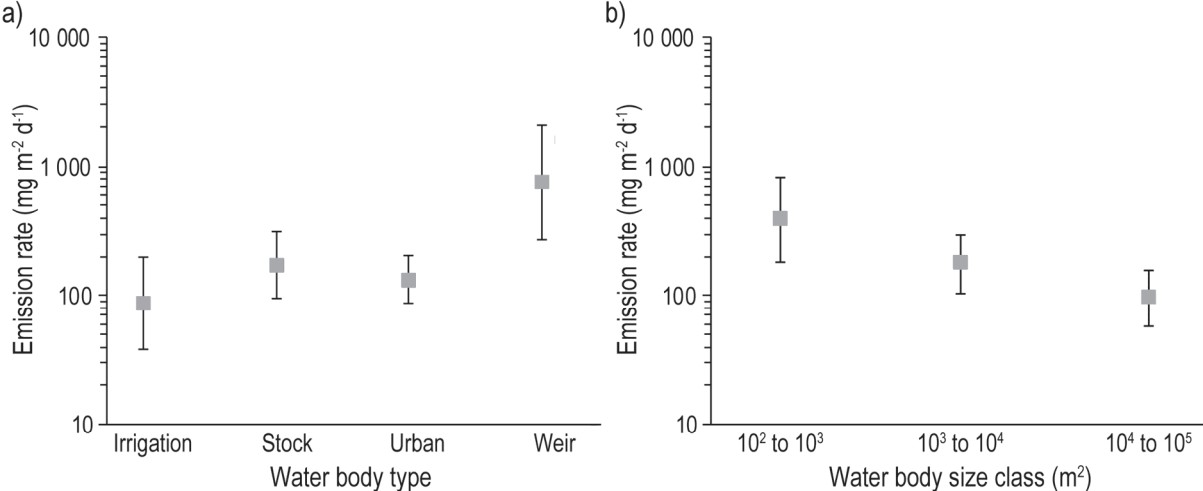

**Figure 3. Mean $CH_4$ emissions across a) four categories of small water bodies (irrigation dams, stock dams, urban lakes and weirs) and b) three GRanD water body size classes. Values indicate geometric mean emission rates and 95% confidence intervals (± 95% CI).**

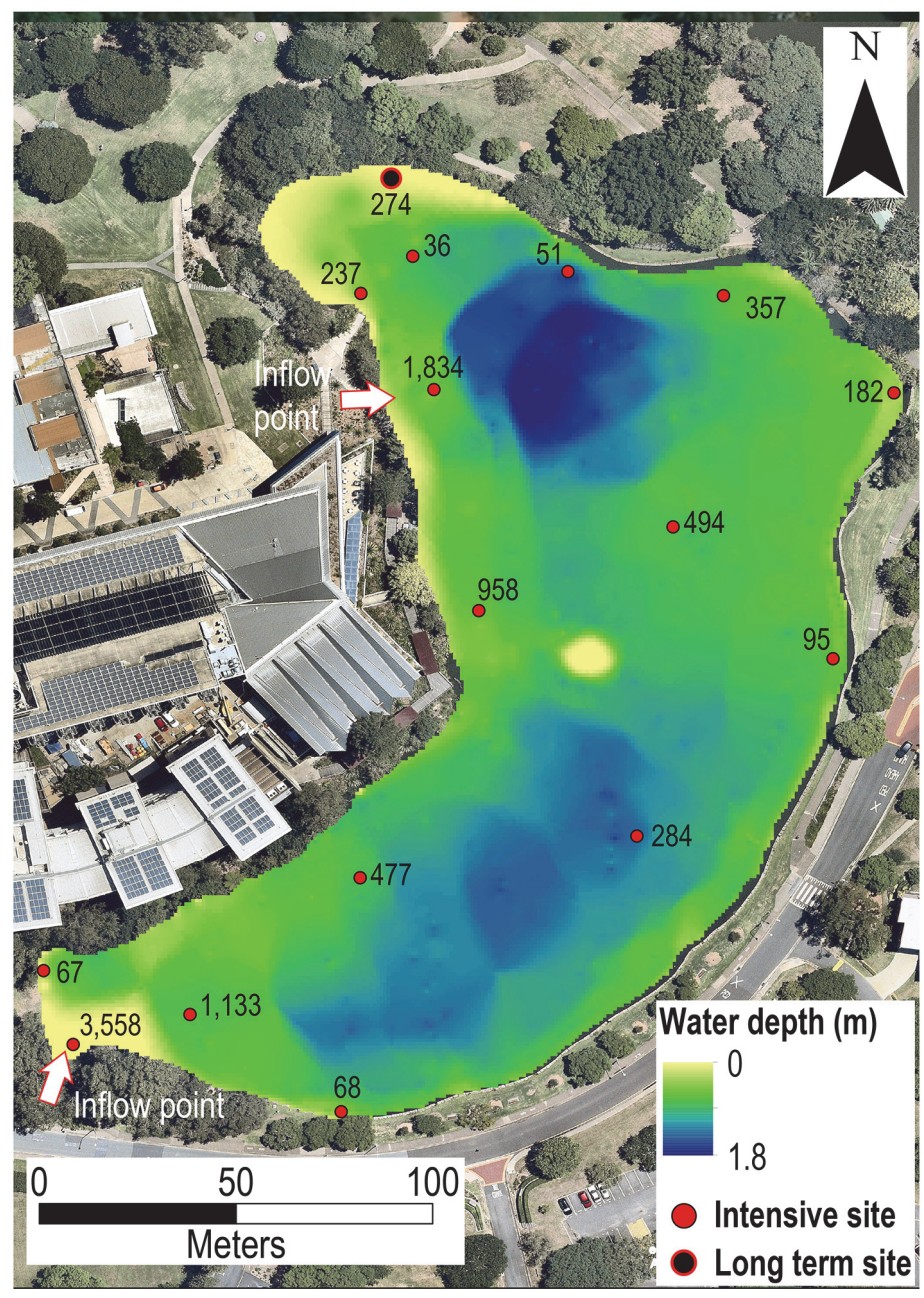

**Figure 4. Sampling site location and chamber emission rates (mg m$^{-2}$ d$^{-1}$) across an urban lake (St Lucia 1) relative to water depth and proximity to stormwater inflow points.**

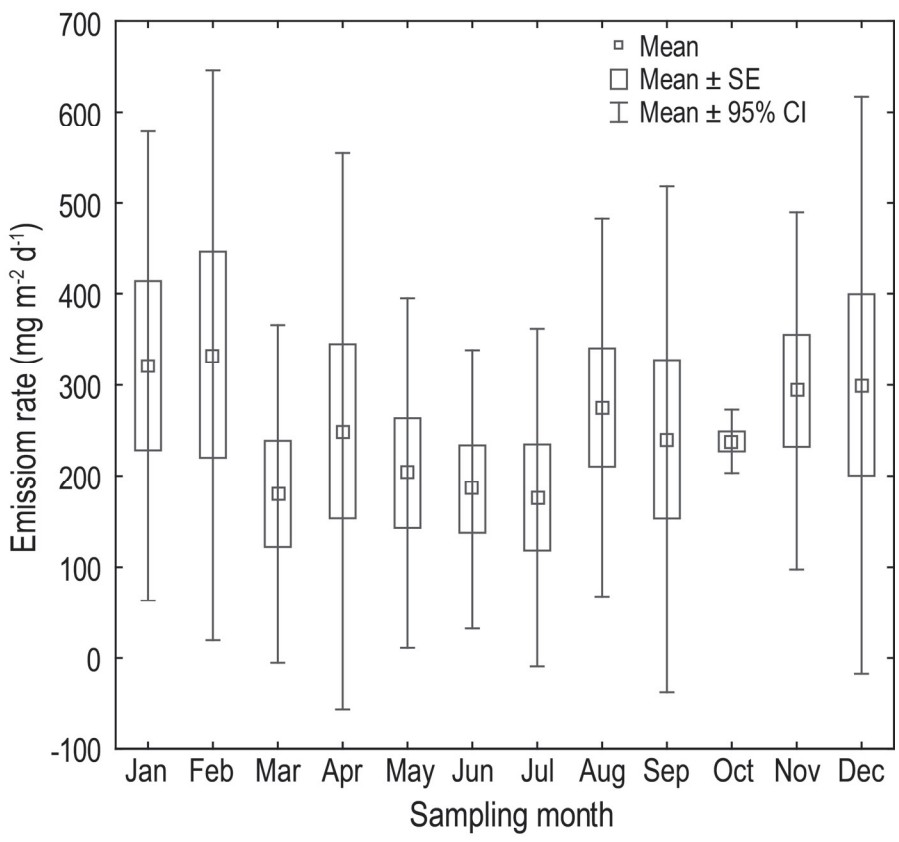

**Figure 5. Monthly CH₄ emissions from a single monitoring site on an urban lake (St Lucia 1) across the annual cycle. Values indicate mean emission rates ± SE (standard error) and 95% CI (confidence intervals).**

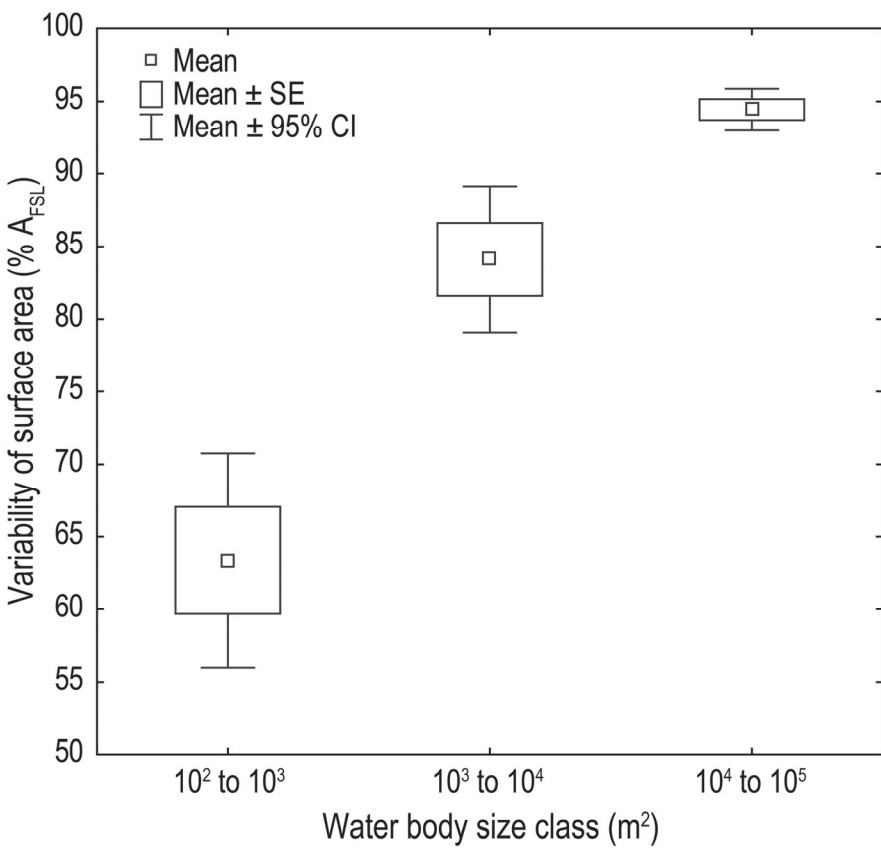

**Figure 6. Variability in water surface area as a percentage of A$_{FSL}$ between three GRanD database size classes of ponds. Values indicate mean surface area ± SE (standard error) and 95% CI (confidence intervals).**

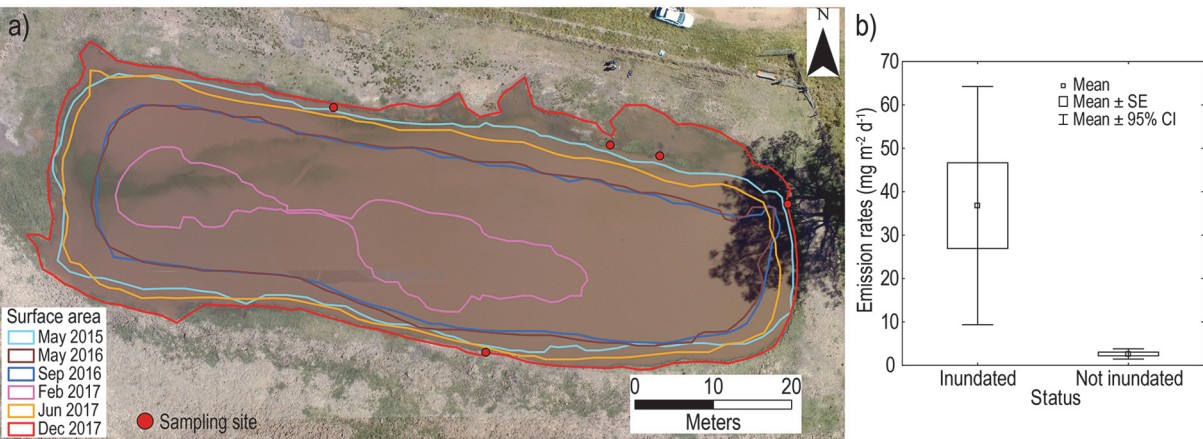

**Figure 7. a) Changes in surface area of stock dam (Gatton 4) over a 40 month period. b) Emissions rates from peripheral zones during a period of inundation and no inundation. Values indicate mean emission rate ± SE (standard error) and 95% CI (confidence intervals).**

**Appendix**

**Table A1: Selected characteristics from individual ponds showing: primary use of each system; surrounding land use type; location of system latitude (Lat) and longitude (Long); average surface area (SA) in m$^2$; mean, median, minimum (Min) and maximum (Max) methane emission rates (mg m$^{-2}$ d$^{-1}$); number of chamber measurements on individual systems (Cham). Primary uses included the following: irrigation for cropping; stock watering for cattle and horses; urban uses included stormwater management and aesthetic purposes; weirs for water supply and stream flow monitoring. * indicates water bodies where repeat sampling was conducted; # indicates water bodies where deployments of less than 24 hours were conducted.**

| Area | Primary use | Land Use | Lat | Long | SA | Arth Mean | Geo Mean | Median | Min | Max | Cham |
|---|---|---|---|---|---|---|---|---|---|---|---|
| Gatton 1* | Irrigation | Grazing | -27.5541 | 152.3412 | 25,903 | 785 | 590 | 527 | 238 | 1,648 | 6 |
| Gatton 2* | Irrigation | Grazing | -27.5548 | 152.3394 | 3,450 | 581 | 170 | 140 | 17 | 2,261 | 6 |
| Gatton 3* | Stock | Grazing | -27.5615 | 152.3434 | 1,041 | 1,149 | 905 | 980 | 314 | 2,007 | 12 |
| Gatton 4* | Stock | Grazing | -27.5625 | 152.3447 | 1,893 | 63 | 55 | 63 | 20 | 109 | 6 |
| Gatton 5 | Irrigation | Cropland | -27.5537 | 152.3503 | 30,458 | 129 | 122 | 110 | 89 | 186 | 3 |
| Gatton 6 | Stock | Cropland | -27.5546 | 152.3488 | 446 | 1,229 | 724 | 844 | 93 | 3,635 | 6 |
| Port precinct# | Urban | Settlement | -27.3917 | 153.1676 | 38,285 | 144 | 57 | 68 | 8 | 357 | 3 |
| St Lucia 1* | Urban | Settlement | -27.4996 | 153.0163 | 22,727 | 632 | 282 | 279 | 36 | 3,558 | 16 |
| St Lucia 2 | Urban | Settlement | -27.4984 | 153.0173 | 4,291 | 92 | 83 | 76 | 51 | 148 | 3 |
| St Lucia 3 | Urban | Settlement | -27.4981 | 153.0167 | 1,755 | 56 | 49 | 43 | 27 | 115 | 5 |
| Pinjarra 1* | Irrigation | Grazing | -27.5372 | 152.9139 | 56,782 | 34 | 15 | 20 | 2 | 122 | 10 |
| Pinjarra 2 | Stock | Grazing | -27.5294 | 152.9242 | 1,943 | 205 | 59 | 277 | 2 | 335 | 3 |
| Pinjarra 3 | Stock | Grazing | -27.5294 | 152.9227 | 210 | 193 | 143 | 107 | 67 | 404 | 3 |
| Oxenford | Urban | Settlement | -27.8924 | 153.2997 | 36,938 | 97 | 94 | 81 | 76 | 133 | 6 |
| Mt Larcom 1 | Stock | Grazing | -23.8008 | 150.9558 | 5,025 | 574 | 37 | 18 | 1 | 2,051 | 5 |
| Mt Larcom 2 | Stock | Grazing | -23.806 | 150.9574 | 1,256 | 48 | 45 | 49 | 26 | 70 | 3 |
| Mt Larcom 3 | Stock | Grazing | -23.8015 | 150.9446 | 16,093 | 17 | 17 | 18 | 14 | 19 | 3 |
| Fig Tree Park | Urban | Settlement | -27.5394 | 152.9682 | 8,357 | 709 | 301 | 289 | 19 | 1,850 | 5 |
| Greenbank# | Stock | Settlement | -27.7249 | 152.9779 | 575 | 290 | 166 | 188 | 29 | 755 | 4 |
| Lake Alford# | Urban | Settlement | -26.2152 | 152.6848 | 21,689 | 49 | 29 | 62 | 5 | 79 | 3 |
| Mt Cootha* | Weir | Forest | -27.4763 | 152.9642 | 580 | 2,493 | 1,405 | 2,337 | 368 | 5,425 | 6 |
| Indooroopilly | Weir | Settlement | -27.5027 | 152.988 | 436 | 413 | 274 | 314 | 77 | 947 | 4 |

**Table A2: Surface area (SA) of Queensland artificial water bodies within each GRanD database size class showing the official land use assessment estimates (QLUMP, 2018) and the revised estimates for the smallest three size classes found in this study.**

| GRanD size class (m$^2$) | QLUMP SA (km$^2$) | Revised SA (km$^2$) |
|---|---|---|
| $10^2$ to $10^3$ | 0.005 | 50.3 |
| $10^3$ to $10^4$ | 8.4 | 400 |
| $10^4$ to $10^5$ | 459 | 637 |
| $10^5$ to $10^6$ | 605 | 605 |
| $10^6$ to $10^7$ | 555 | 555 |
| $10^7$ to $10^8$ | 553 | 553 |
| $10^8$ to $10^9$ | 448 | 448 |
| **Total** | **2,629** | **3,248** |

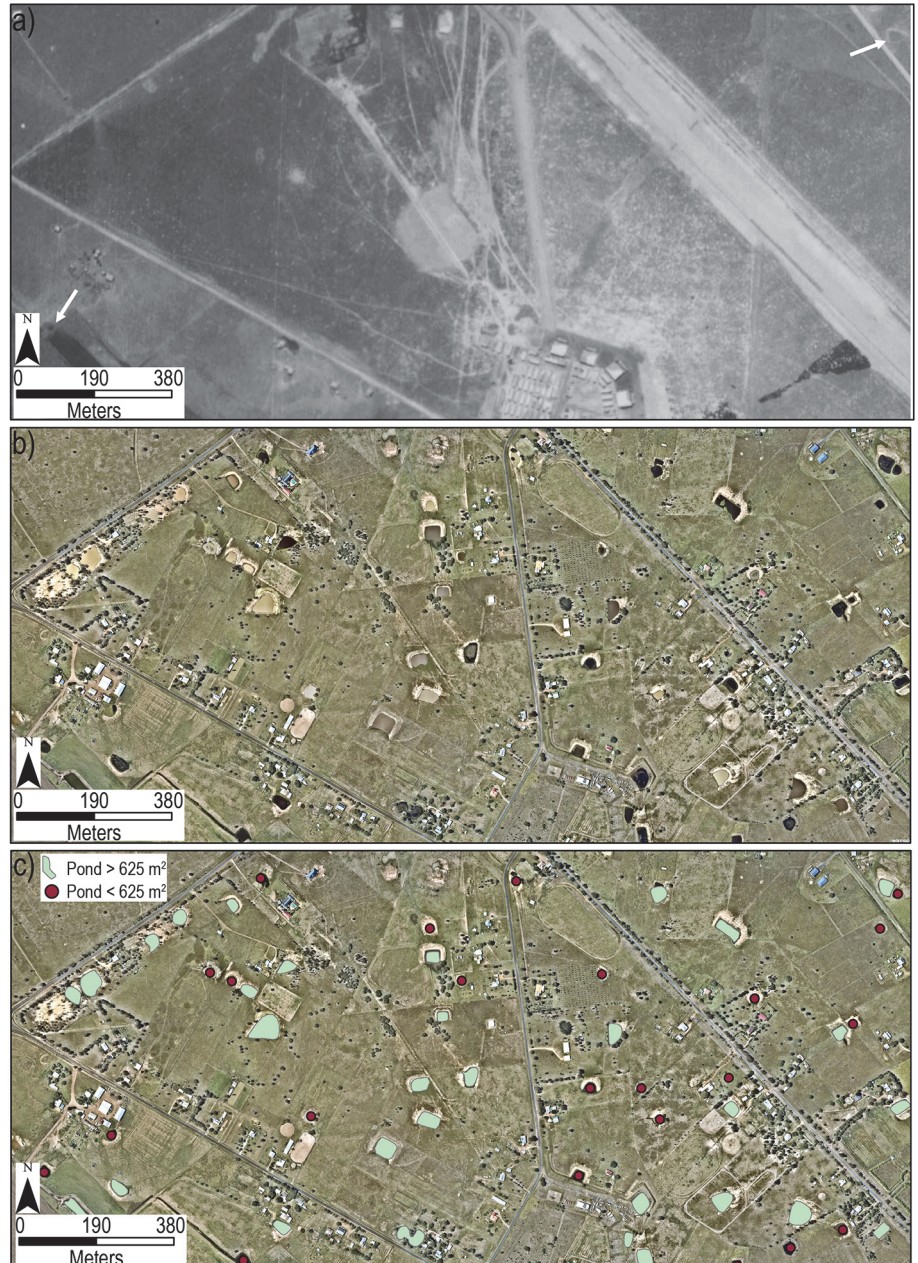

**Figure A1. Historical changes in pond distribution from a 2.7 km$^2$ area in south east Queensland, Mt Tarampa (27°27'44"S, 152°28'59"E). a) 1944 aerial images showing 2 ponds indicated by white arrows, b) 2017 aerial image showing 54 ponds and c) showing the relative distribution of ponds from Reservoir (>625 m$^2$) database and Water Storage Point (<625 m$^2$) database, and together this results in a density of 20 ponds km$^{-2}$.**

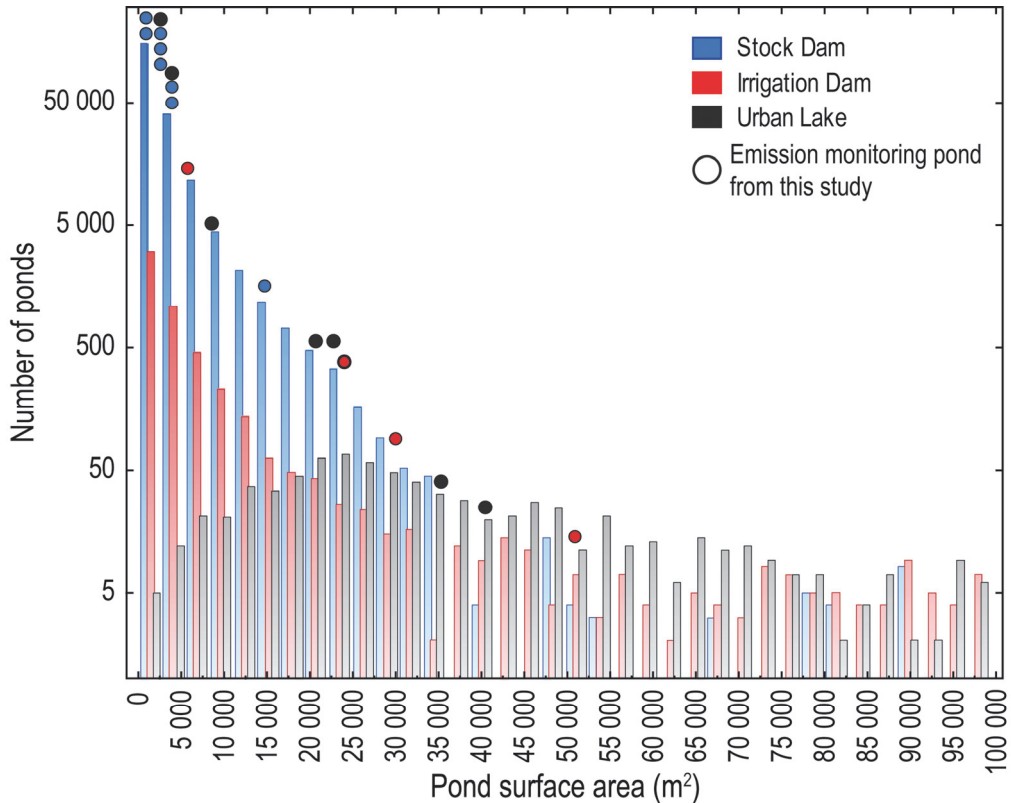

**Figure A2. Pond size from emission study relative to histogram of regional pond distribution of stock dams, irrigation dams and urban lakes. The surface area of pond used emission study (Table A1). Histogram of regional distribution of ponds was developed from QLUMP, Reservoir and Water Storage Points databases and separated into pond type depending on surrounding land use: "Grazing native vegetation" for stock dams; "Production from irrigated agriculture and plantations" for irrigation dams; "Intensive uses" for urban lakes with "Mining" and "Manufacturing" landuse within "Intensive Uses" were removed to ensure only urban areas were selected. To incorporate the distribution of ponds within the Water Storage Points database, it was assumed this would match the distribution from the 100 individual ponds examined in Section 2.2 to determine their average surface area.**

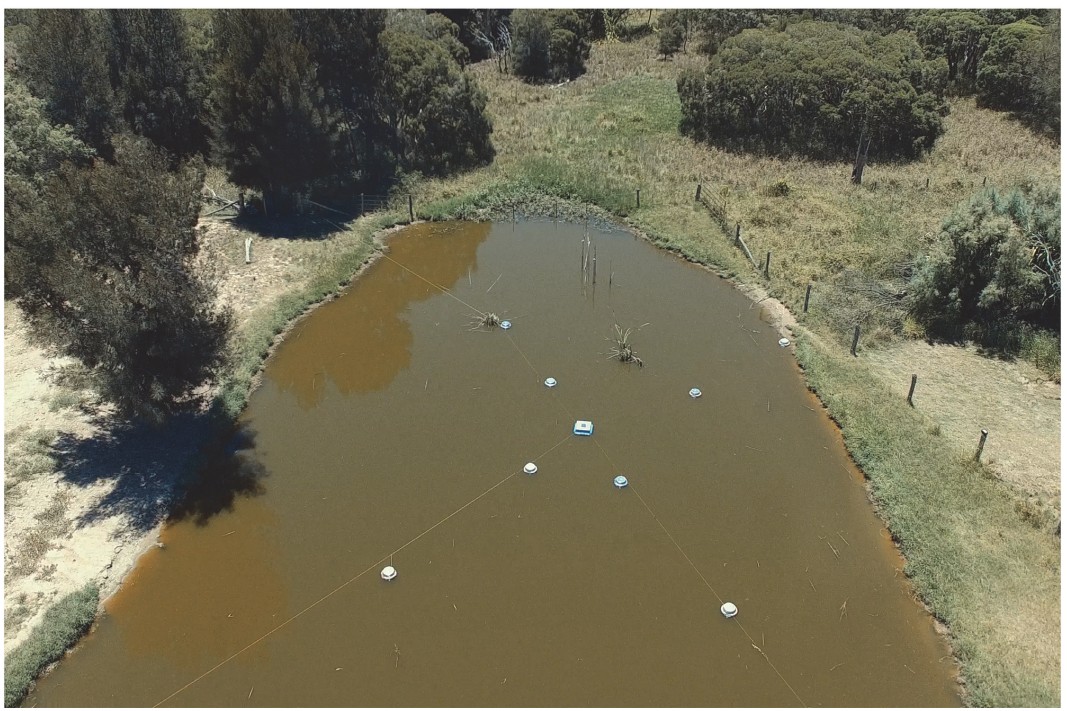

**Figure A3. An oblique drone image showing a nine floating chamber deployment setup targeting peripheral and central zones on a stock watering dam (Gatton 3).**

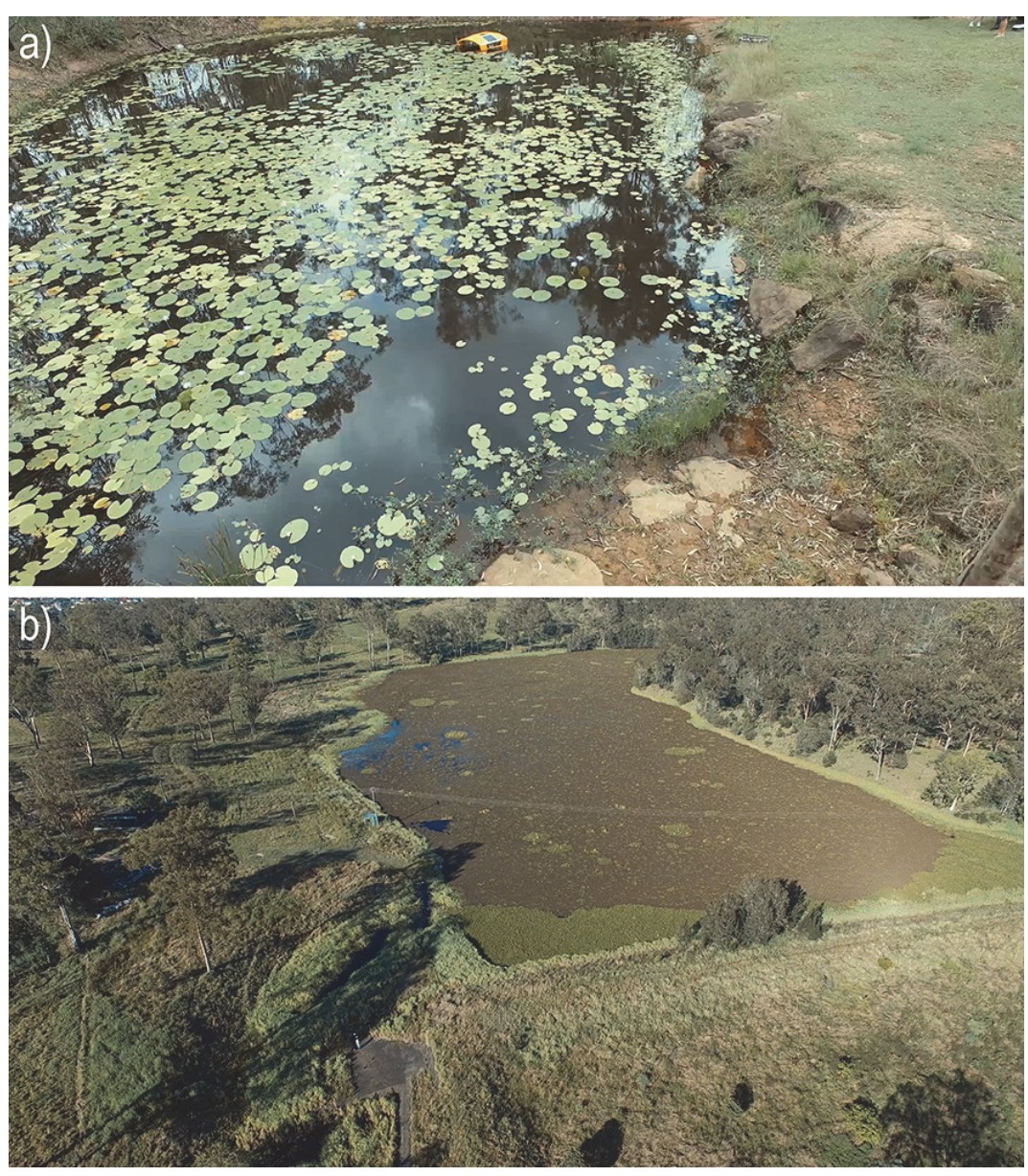

**Figure A4. Oblique drone images showing natural obstacles for pond chamber deployments from a) emergent macrophytes and b) floating aquatic weeds.**

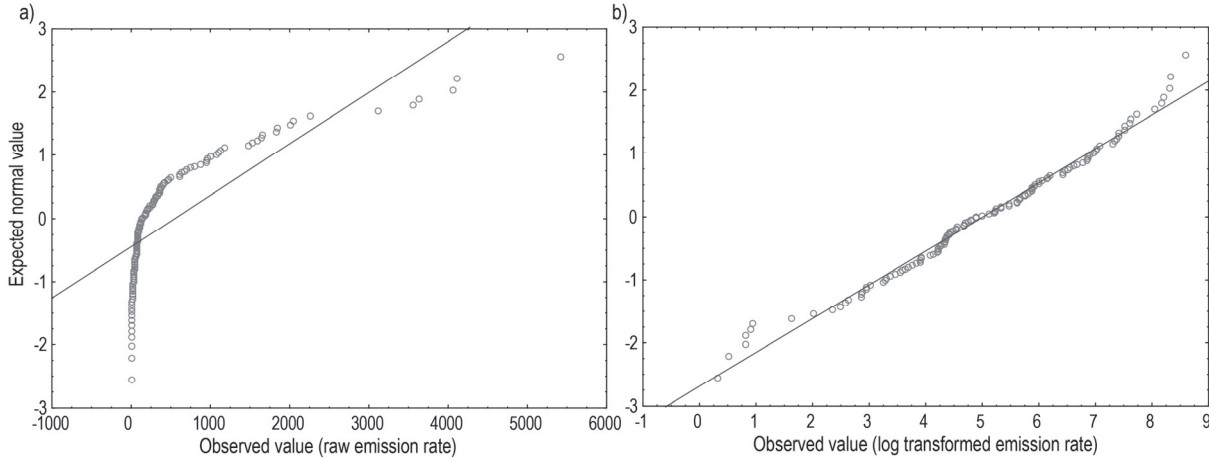

**Figure A5. Normal probability plots for a) raw methane emissions and b) log transformed emissions data. Shapiro-Wilks tests p-value for raw emissions data was < 0.001 and failed the normality test; p-value for log transformed emissions data was 0.081 indicating data was normally distributed.**

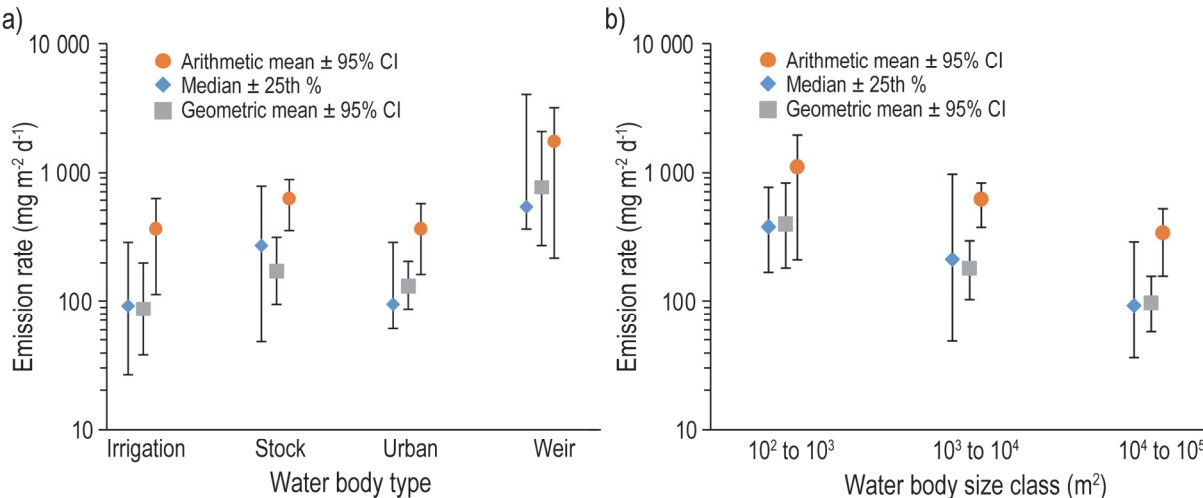

**Figure A6. Three measures of centrality for methane emissions across a) four categories of small water bodies (irrigation dams, stock dams, urban lakes and weirs) and b) three GRanD water body size classes. Error for each measure are as follows: median emission rates and interquartile range (± 25th %), arithmetic and geometric mean emission rates and 95% confidence intervals (± 95% CI).**