# Peer review of "The importance of small artificial water bodies as sources of methane emissions in Queensland, Australia."

_Hydrology and Earth System Sciences, 2018_

## Referee Comment (RC1) · C. Nadia (Referee) · 14 Aug 2018

This manuscript deals with an important issue, namely emissions of the strong greenhouse gas methane from the numerous small ponds in Queensland, Australia, in the context of climate changes and ponds number increase. Assessment of methane emissions supposes two steps: first the survey of small ponds and their relevant characteristics, second the assessment of methane emissions, depending on the pond characteristics (including size, type, location, climate and seasonality).

In this manuscript, the first step is realized based on existing data bases, the second on quite heavy emissions, and the dependence on emissions rate on inundated/non-inundated status of soil, the authors carried out complementary measures to characterize these causes of variability. The observations are then used to extrapolate emissions assessment to the global set of ponds, with two alternative extrapolation methods. This approach seems relevant. Anyway, several choices should have been discussed more in detail, and some underling hypothesis should have been clarified. In its present status, the manuscript is based on a large amount of data, not fully exploited. In particular, the way the "complementary" data is used (or not) is not clear.

Here are some more specific comments illustrating the above comments:

**Introduction:**

Introduction could have also cited the other kinds of greenhouse gases y emitted by artificial ponds (CO2, N2O), and could have evoked their potential role of organic carbon sink. Balancing these two antagonist influences would give a larger context to the rest of the manuscript.

P2 - l40, it would be interesting to give some orders of magnitude of the CH4 sink effect of soils prior inundation, to compare with the emission rates presented later.

P3-l5: it seems a $5^{th}$ point is missing, which is the extrapolation of the four other points results in a regional emission assessment: at this is not straightforward, it should be mentioned. Point 3 is a bit misleading, as spatial and temporal variability in emission rate is in fact assessed only for one pond, independently of the pond's area variation.

**2.1 Study area description**

The link between the fact that the majority of artificial water bodies are less than 5 ML and the choice to study emissions from ponds which area is less than $10^5 m^2$ is not clear. Why this threshold?

**2.2 Relative surface area of ponds across the region**

Given the discrepancy between the different sources of data and the difficulty to identify ponds and their characteristics within a large area, it would have been useful to give more details on the building of these three databases: how is data acquired, which kind of characteristic does each database include, at which periodicity is it revised, …

Given the strong dependency of CH4 emission rate on the ponds' size, why choosing an average area for ponds <625m², instead of using a size distribution. The same question arises for the classification in the three classes of the GRanD. This seems premature before the later results. Anyhow, it would have been interesting to present a histogram of the ponds' sizes. It is also disappointing not to know more about the types of ponds (and the potential link between type and size), their location, the way they are supplied in water, … all characteristic which may influence methane emissions and which may be available in the databases?

**2.3 CH4 emissions from broad spectrum of pond types**

It would have been useful for readers not familiar with methane emission from waterbodies to present the different kind of methane emissions measures, their advantages and limits, and to argument the choice performed here. Above all, the choice of the studied ponds should have been discussed, and their representativeness of the whole ponds set variety assessed.

How was the number of floating chambers per pond chosen? It seems not to be only in function on the pond's size? Was the uncertainty arising from measuring only 6 to 8 hours of emission for 3 ponds assessed? Is the emission process known to be varying at the daily scale? When did the monitoring occurred during the year (and which year rather dry or wet) ?

Nothing is said about the way the pond global emission rate is assessed from the punctual measures: given the variability at the pond scale illustrated with 2.4.1 results, it yet seems crucial. Uncertainty arising from how this calculation was performed may be as high as those arising from the choice of arithmetical or geometrical means between ponds' emission rates later on (2.6).

**2.4 Spatial and temporal variability in surface area and emission rate**

**2.4.1**

Here also it is not clear why this pond was chosen rather than another. What about its representativeness? For example, one can expect that the temporal variability of a weir's emission rate to be higher than an urban's lake one? The year when this monitoring was performed is not specified, neither corresponding rainfall, which may be of influence on the pond supply and the emission processes? Air and water temperature may also be influent factors?

**2.4.2**

Is rainfall variability homogeneous at the state scale? In other words, are the percentages of $A_{FSL}$ calculated for the ponds which were monitored relevant for the regional scaling which is performed in 4.1?

**2.5**

As for the 2.4.1 section, it is not clear why this pond was chosen rather than another. What assures its representativeness of other small ponds which area varies a lot depending on rainfall?

These complementary field campaigns are honest and interesting attempts to deepen the study, but should be more detailed and argued to be totally useful and convincing to strengthen the results which are the main scope of the paper.

**3.2 CH4 emissions from ponds**

Some considerations on ebullition/diffusive emissions would be necessary to support the conclusion that ebullition is the dominant emission pathway. If so, spatial heterogeneity of emission must be high: is the monitoring protocol adapted to capture this heterogeneity?

As said before, a histogram of sizes and types of ponds would be useful to contextualize the results. Detail of the way the emissions are assessed at the pond scale too.

As weirs present high emission rates compared to other types of ponds, is it relevant to group them with other small ponds/stock ponds to assess the regional scale emissions?

**3.3.1 Spatial and temporal variability within a single pond**

These data are no doubt very interesting, but what they bring to this study is not clear for me. How are they used to the following regional scaling? If it is by considering that observed emissions from the other ponds can be taken as annual averages, this should be clarified. If this is the case, the validity of this hypothesis should be discussed, as only one type of pond was considered for this analysis.

It seems to me that this part could be cut from this manuscript, and maybe give the material for another paper, which would allow to analyse more in depth the emissions variability and the factor which influence it.

**4.1**

In India, numerous ponds are used to increase groundwater discharges. As a consequence, $CH_4$ emission from these ponds may differ from the types of ponds which were studied here. This could be specified to be rigorous in this discussion about the importance to take into account ponds in methane emission rates assessment.

**4.2**

Again, more details should have been given above on the different emission pathways and their influencing factors.

As depth, the way water is supplied in pond and substrate seem to be influential, this discussion could address the question of the availability of such data in current databases.

**5 Future research**

Some sources of uncertainties are taken into account in this manuscript, other are not: this section could emphasize these points, and discuss how to handle them (assessment of a whole pond emission rate given punctual data in time and space; identification of emission pathways, characterisation of the way some factors influence emission rates–type, depth, purpose, water supply, temperature …) . At the moment, the research perspectives are more or less an extension of the work which was already performed.

---

## Referee Comment (RC2) · Anonymous Referee #2 · 20 Aug 2018

The importance of small artificial water bodies as sources of methane emissions in Queensland, Australia 1. Does the paper address relevant scientific questions within the scope of HESS? Yes 2. Does the paper present novel concepts, ideas, tools, or data? Yes 3. Are substantial conclusions reached? Yes 4. Are the scientific methods and assumptions valid and clearly outlined? No 5. Are the results sufficient to support the interpretations and conclusions? Yes 6. Is the description of experiments and calculations sufficiently complete and precise to allow their reproduction by fellow scientists (traceability of results)? No, more details are needed on the chamber method. 7. Do the authors give proper credit to related work and clearly indicate their own new/original contribution? Yes 8. Does the title clearly reflect the contents of the

paper? Yes 9. Does the abstract provide a concise and complete summary? Yes 10. Is the overall presentation well structured and clear? Could use some improvement. 11. Is the language fluent and precise? Yes, with a few exceptions in the discussion and conclusions. 12. Are mathematical formulae, symbols, abbreviations, and units correctly defined and used? Yes, with the one exception of units for CH4 flux which needs clarification. 13. Should any parts of the paper (text, formulae, figures, tables) be clarified, reduced, combined, or eliminated? The methods need some clarification; the conclusions need to be distilled. 14. Are the number and quality of references appropriate? Yes 15. Is the amount and quality of supplementary material appropriate? Yes

Summary: The global importance of GHG emissions from small ponds is not well understood due to a lack of knowledge of both their 1) GHG emission rates and 2) cumulative spatial extent. This study quantifies both for the state of Queensland, Australia, and the authors also propose two upscaling approaches. In addition, the study investigates spatial (both intra- and inter-pond), temporal variability in emissions, and the impact of variable inundation status on emissions. The study found that including small ponds in the calculation of total surface area of artificial water bodies increased the cumulative total surface area by 24%. Spatial variability within ponds was found to be much greater than temporal variability.

Overarching comments: This study is an important contribution to the literature on methane emissions from inland waters, and well executed. My main comment is that the manuscript could benefit from some clarification and reorganization. There are several additions needed in the methods, several items stated in results that should be included in the methods, several items that belong in the results that are in the discussion (see specific comments, below), and more attention to study components 2-4 (laid out in the introduction) in the discussion. Methods: • Several sections in the methods could be clearer if the intention was stated in a topic sentence leading each paragraph. For example, in section 2.2 the onus is put on the reader to figure

out what parts are for the purpose of determining individual water body sizes, what parts are for determining the cumulative area of small ponds, and what parts are for determining the size distribution. • In section 2.4, the total number of sample ponds should be stated, as should the method for choosing the subset that purportedly represents the wide spectrum of ponds • More detail on the chamber method should be included. Concerns about the methodology that need to be addressed include: o Biases in the emission measurement due to diffusive uptake of methane from the chamber headspace to the water under conditions of high methane partial pressure in the chamber headspace Discussion: • The introduction lays out four components of the study, but the discussion is heavily weighted to component 1: "Quantify the area of ponds, relative to regional assessments of larger artificial water bodies", and to unstated components/objectives of scaling. Either: o the introduction should be revised to reflect the structure of the discussion, e.g. state more clearly how components 2-4 support upscaling of inland water emission estimates, why determining the pathway is important or o the discussion should be revised to address components 2-4

Specific Comments: • Page 2 lines 2-3 introduce the concept of uncertainty in surface area and classes of artificial water bodies. It is unclear what the authors mean by this – differences in how water bodies are classified? Different classification schemes? This sentence cites: o Surface area: Chumchal et al., 2016 Abundance and size distribution of permanent and temporary farm ponds in the southeastern Great Plains, Inland Waters o Classes: Panneer Selvam et al., 2014 Methane and carbon dioxide emissions from inland waters in India–implications for large scale greenhouse gas balances • Page 2 line 15: correct typo "the creation of water small artificial water bodies" • Page 2 line 18: "these can be considered anthropogenic in origin" the use of "can" makes this statement sound like it is optional or up to someone's discretion to categorize emissions from flooded lands as anthropogenic or natural. Clarify this sentence by restating as "these emissions are considered anthropogenic in origin according to [IPCC guidelines], and should therefore be. . ." • Page 2, lines 26-28: the parenthetical statement is distracting. It seems the authors wrote it this way because

the goal "to determine the factors that account for spatial and temporal variability in the flux" is a sub-goal of obtaining CH4 flux measurements from a broader range of sites. I recommend moving the parenthetical statement to a sentence following this one: "An important part of the value of building a dataset of CH4 flux estimates from a broad range of sites is determining factors that account for spatial and temporal variability in the flux." • Page 3, line 6: this is the first mention of inundation level influencing emission rates. This idea should be introduced in the introduction; the intro as it is currently just deals with the difficulty of estimating total surface area due to changing surface areas • Page 3, line 9: change "having" to "has" • Page 3, line 10 – 11: stating that 80% of the land is used for agriculture could be supported by figure 2, as could the statement about rainfall gradients on lines 14-15. • Figure A1: where are the two ponds in panel a)? Could they be pointed to with arrows, for example? • Page 4, line 8-9: How was the mean surface area used to calculate the total surface area? Please provide an equation, or at least spell it out with more clarity. Scaled simply as mean size per pond * total number of ponds? Any size binning or other weighting? It is unclear if the sentences following are clarifications on the surface area determination methodology, or are background information for the two upscaling approaches • Figure 2: add the Category titles to the legend in panel a). For b) and c), increase the font size. Is it possible to indicate the location of the 22 study lakes on this map? • Page 4, line 25: please clarify if several emission measurements were taken over 24-hour periods, or if the chamber incubation period was 24 hours. How many headspace gas samples were taken per emission measurement? Also, what time frame were these measurements made over? • Page 5, line 4: mention the number of ponds monitored, i.e. change to "The variability in surface area of each of the 22 ponds monitored in the emissions surveys was analyzed. . ." • Page 6, line 16: is it possible to provide more quantitative evidence than "clearly" for the lognormal fit? In the Figure caption, a p-value is mentioned, but not in the text. • Page 6, line 28: what additional datasets? • Page 6, line 35-36: Your first research objective was to quantify the area of ponds relative to regional assessments of larger artificial water

bodies – how does the 1,000 km2 compare to the total artificial water body surface area in Queensland? • Page 7, line 3: change mg m-2 d-1 to mg CH4 m-2 d-1 or mg CH4-C m-2 d-1 depending on which you mean. I'm guessing the former, but the latter is also sometimes used. • Page 8, line 13: remove "clearly" • Page 8, line 14-15, 19-20: these results should be moved to section 3.1; however, the discussion of their importance is appropriate to have here • Page 8, like 20-22: how do these emissions compare to mean annual CH4 emissions from larger inland waters in the state? • Page 9, line 36: what do you mean by "available for emissions"? This phrasing is unclear. Consider changing to "as this will greatly improve the surface area estimate of flooded lands used for upscaling greenhouse gas emissions."

---

## Author Comment (AC2) · 25 Sep 2018

**General Comments:**

*Summary: The global importance of GHG emissions from small ponds is not well understood due to a lack of knowledge of both their 1) GHG emission rates and 2) cumulative spatial extent. This study quantifies both for the state of Queensland, Australia, and the authors also propose two upscaling approaches. In addition, the study investigates spatial (both intra- and inter-pond), temporal variability in emissions, and the impact of variable inundation status on emissions. The study found that including small ponds in the calculation of total surface area of artificial water bodies increased the cumulative total surface area by 24%. Spatial variability within ponds was found to be much greater than temporal variability.*

*Overarching comments: This study is an important contribution to the literature on methane emissions from inland waters, and well executed.*

*My main comment is that the manuscript could benefit from some clarification and reorganization. There are several additions needed in the methods, several items stated in results that should be included in the methods, several items that belong in the results that are in the discussion (see specific comments, below), and more attention to study components 2-4 (laid out in the introduction) in the discussion.*

**Reply:** Thank you for all these comments and suggestions and we have addressed them all which has greatly improved the manuscript. Detailed responses are provided below. Page and Line numbers refer to the attached revised manuscript which contains track changes for changes suggested by both reviewers.

*Methods: Several sections in the methods could be clearer if the intention was stated in a topic sentence leading each paragraph. For example, in section 2.2 the onus is put on the reader to figure out what parts are for the purpose of determining individual water body sizes, what parts are for determining the cumulative area of small ponds, and what parts are for determining the size distribution.*

**Reply:** We have rephrased the leading sentences of Methods sections to clarify the intention of each component. The revised text can be found in the following sections:

P4 L21-24:

"To determine the number and relative surface area of ponds across the Queensland, three State Government GIS databases of artificial water bodies were utilised. However these databases required additional processing to extract comparable pond data as there were inconsistencies in the format and nomenclature of feature types."

P6 L14-16:

"To gain insight into the spatial and temporal uncertainty in pond emissions we compared variability in seasonal emissions from a single site to emissions from an intensive spatial survey of multiple sites across the pond (Fig. 4)."

P7 L5-6:

"This pond was selected as stock dams generally experience accelerated rates of water level change due to their relatively small size compared to other pond types (Fig. A2)."

*In section 2.4, the total number of sample ponds should be stated, as should the method for choosing the subset that purportedly represents the wide spectrum of ponds*

**Reply:** We have now stated the total number of sample ponds P6 L28:

"The variability in surface area of each of the 22 ponds monitored in the emissions surveys was analysed"

We have provided a histogram of pond size for the major pond types across the region and included the sample ponds on this histogram to support their representativeness of the regional spectrum of ponds. Appendix P30 Figure A2:

[Figure]

**Figure A2. Pond size from emission study relative to histogram of regional pond distribution of stock dams, irrigation dams and urban lakes. The surface area of pond used emission study (Table A1). Histogram of regional distribution of ponds was developed from QLUMP, Reservoir and Water Storage Points databases and separated into pond type depending on surrounding land use: "Grazing native vegetation" for stock dams; "Production from irrigated agriculture and plantations" for irrigation dams; "Intensive uses" for urban lakes with "Mining" and "Manufacturing" landuse within "Intensive Uses" were removed to ensure only urban areas were selected. To incorporate the distribution of ponds within the Water Storage Points database, it was assumed this would match the distribution from the 100 individual ponds examined in Section 2.2 to determine their average surface area.**

*More detail on the chamber method should be included. Concerns about the methodology that need to be addressed include: Biases in the emission measurement due to diffusive uptake of methane from the chamber headspace to the water under conditions of high methane partial pressure in the chamber headspace*

**Reply:** We have addressed this oversight in the methodology description and have provided a more detailed description to address these concerns. The following text and supporting references have been included in the manuscript P5 L27-33:

"There are a number of commonly used methods to assess methane emissions from water bodies depending on the pathway of interest. For the diffusive emission pathway, rates may be modelled using the thin boundary methods or directly measured using manual or automatic floating chambers (St. Louis et al., 2000). For ebullition pathways, rates can be directly measured using acoustic surveys or funnel traps (DelSontro et al., 2011). Thin boundary layer models cannot be used to quantify the ebullition pathway and acoustic surveys or funnel traps cannot be used effectively in ponds as the water depth is often too shallow (< 1 m). We chose to use floating chambers to capture both ebullition and diffusive fluxes."

And P5 L37-39:

"The floating chambers used were designed to yield negligible bias on the gas exchange and compare well with non-invasive approaches (Cole et al., 2010;Gålfalk et al., 2013;Lorke et al., 2015)."

And P6 L2-8:

"The 24 hour deployment time was chosen to increase the likelihood to capture ebullition, which is episodic in nature, and to incorporate diel variability in diffusive emissions which can be up to a 2-fold bias (Bastviken et al., 2004;Bastviken et al., 2010;Natchimuthu et al., 2014). The use of long term deployments may underestimate diffusive fluxes, which decrease as the chamber headspace approaches equilibrium with the water. However, in contrast to $CO_2$, $CH_4$ has a long equilibration time and it has been shown that a 24 hour deployment of these types of flux chambers on lakes underestimate diffusive fluxes by less than 10% (Bastviken et al., 2010)."

*Discussion: The introduction lays out four components of the study, but the discussion is heavily weighted to component 1: "Quantify the area of ponds, relative to regional assessments of larger artificial water bodies", and to un-stated components/objectives of scaling. Either: o the introduction should be revised to reflect the structure of the discussion, e.g. state more clearly how components 2-4 support upscaling of inland water emission estimates, why determining the pathway is important or o the discussion should be revised to address components 2-4*

**Reply:** We have now revised the objectives to clarify the links between the four components and the integration of these for the regional assessment. The new text can be found P 3 L19-28:

"The principle objective of this study was to establish the GHG status of ponds in Queensland, Australia. Given the paucity of GHG data from ponds, this study has focussed on empirical assessments of $CH_4$ emissions from a range of pond types rather than detailed assessments of drivers of these emissions. Our assessment comprised four components:

1. Quantify the area of ponds, relative to regional assessments of larger artificial water bodies;
2. Quantify $CH_4$ emission rates for a wide spectrum of pond types;
3. Determine variability in their surface area and emission rates;
4. Determine the influence of inundation level on emission rates.

When integrated together, these components provide a robust regional assessment of anthropogenic $CH_4$ emissions for ponds in Queensland, Australia."

In addition we highlighted the importance of the regional upscaling relative to larger water bodies in the discussion P10 L12-14:

"Given ponds represent 33.5% of the total flooded lands surface area in Queensland and emission rates are equivalent to larger water bodies in the region (Musenze et al., 2014;Sturm et al., 2014), ponds represent one-third of total emissions from flooded lands in Queensland."

**Specific Comments:**

*Page 2 lines 2-3 introduce the concept of uncertainty in surface area and classes of artificial water bodies. It is unclear what the authors mean by this – differences in how water bodies are classified? Different classification schemes? This sentence cites: o Surface area: Chumchal et al., 2016 Abundance and size distribution of permanent and temporary farm ponds in the southeastern Great Plains, Inland Waters Classes: Panneer Selvam et al., 2014 Methane and carbon dioxide emissions from inland waters in India–implications for large scale greenhouse gas balances*

**Reply:** We have revised this section of the introduction to be more explicit about these sources of uncertainty and the site description was revised to highlight issues in classifying artificial water bodies. P2 L6-7:

"Furthermore the surface area of small water bodies can be particularly difficult to quantify in national and global datasets due to their small size and large number (Chumchal et al., 2016)."

And P4 L8-15:

"However, the number and surface area of ponds in Queensland is relatively unknown as there is no legal requirement to refer ponds to the state registry due to their small size.  Under current state law only dam walls in excess of 10 m and volumes above 750 ML are referable (DEWS, 2017) and the maximum reported volume for ponds in Queensland is three times less than the referable volume (< 250 ML) (SKM, 2012). This study has assumed ponds are less than 100,000 $m^2$ as this is recognised globally as the major area of uncertainty in surface area assessments (Lehner and Doll, 2004;Downing 2010) and has been identified as a threshold in global lake inventories (Downing et al., 2006;Verpoorter et al., 2014)."

*Page 2 line 15: correct typo "the creation of water small artificial water bodies"*

**Reply:** We have corrected this typo P2 L24.

*Page 2 line 18: "these can be considered anthropogenic in origin" the use of "can" makes this statement sound like it is optional or up to someone's discretion to categorize emissions from flooded lands as anthropogenic or natural. Clarify this sentence by restating as "these emissions are considered anthropogenic in origin according to [IPCC guidelines], and should therefore be. . ."*

**Reply:** We have revised this statement as suggested P2 L27:

"these emissions are considered anthropogenic in origin according to IPCC guidelines (IPCC, 2006)"

*Page 2, lines 26-28: the parenthetical statement is distracting. It seems the authors wrote it this way because the goal "to determine the factors that account for spatial and temporal variability in the flux" is a sub-goal of obtaining CH4 flux measurements from a broader range of sites. I recommend moving the parenthetical statement to a sentence following this one: "An important part of the value of building a dataset of CH4 flux estimates from a broad range of sites is determining factors that account for spatial and temporal variability in the flux."*

**Reply:** Thank you for the recommendation, we have moved the parenthetical statement as suggested P3 L3-6:

"An important part of the value of building a dataset of CH4 flux estimates from a broad range of sites is determining factors that account for spatial and temporal variability in the flux."

*Page 3, line 6: this is the first mention of inundation level influencing emission rates. This idea should be introduced in the introduction; the intro as it is currently just deals with the difficulty of estimating total surface area due to changing surface areas*

**Reply:** We have now introduced the effect of inundation level on emission rates in the Introduction P2 L8-12:

"In addition, the peripheral areas of small water bodies regularly experience periods of inundation and no inundation as water levels change due to their relatively shallow nature and high water use rates. The changes in their inundation status may influence emission rates as has been observed for natural ponds (Boon et al 1997)."

*Page 3, line 9: change "having" to "has"*

**Reply:** We have changed this typo P3 L37.

*Page 3, line 10 – 11: stating that 80% of the land is used for agriculture could be supported by figure 2, as could the statement about rainfall gradients on lines 14-15.*

**Reply:** We now support these statements with agricultural land use Figure 2 a (P3 L39) and rainfall gradients Figure 2 b (P4 L4).

*Figure A1: where are the two ponds in panel a)? Could they be pointed to with arrows, for example?*

**Reply:** Two white arrows have now been added to the figure to point out the ponds Appendix P29 Figure A1:

[Figure]

The figure legend has additional explanatory text P29 L3-4:

"**1944 aerial images showing 2 ponds indicated by white arrows**"

*Page 4, line 8-9: How was the mean surface area used to calculate the total surface area? Please provide an equation, or at least spell it out with more clarity. Scaled simply as mean size per pond \* total number of ponds? Any size binning or other weighting? It is unclear if the sentences following are clarifications on the surface area determination methodology, or are background information for the two upscaling approaches.*

**Reply:** We have provided a more detailed explanation for the total surface area calculation P5 L7-13:

"…then assumed to approximate the surface area of all individual ponds within this database and the total surface area was calculated by multiplying this mean surface area by the total number of ponds.

To ensure only one water body was reported from each location, all databases were first screened to remove repeated detections of waterbodies. All remaining water bodies were then summed together to calculate total surface area of ponds and this was compared to larger reservoirs to determine their relative surface area."

To scale the pond surface we used the mean surface area multiplied by the total number of ponds as the relationship between pond size and emission rate was too weak to support weighting the emissions within this pond size class:

[Figure]

We have changed the following sentences to clarify the upscaling approach taken P5 L12-14:

"All remaining water bodies were then summed together to calculate total surface area of ponds and this was compared to larger reservoirs to determine their relative surface area. To undertake regional scaling of pond emissions, individual ponds were sorted using two different size class classifications:"

*Figure 2: add the Category titles to the legend in panel a). For b) and c), increase the font size. Is it possible to indicate the location of the 22 study lakes on this map?*

**Reply:** We have made the suggested changes to this figure and included the study site locations P20 Figure 2:

[Figure]

We have revised the figure legend accordingly P20 L3-11:

"**Figure 2. a) 2018 state wide assessment showing the relative surface area occupied by secondary land use categories (QLUMP, 2018). Note the legend shows the two largest land uses within each category. b) Mean annual rainfall isohyets across Queensland from 30 period of 1961 to 1990 (http://www.bom.gov.au accessed March 2018). c) Location of study ponds and ponds identified from the land use assessment (QLUMP 2018) and two additional state wide databases (see text).**"

*Page 4, line 25: please clarify if several emission measurements were taken over 24-hour periods, or if the chamber incubation period was 24 hours. How many headspace gas samples were taken per emission measurement? Also, what time frame were these measurements made over?*

**Reply:** Measurements were taken at chamber deployment and a final sample after 24 hours, this is now included in the text as well an explanation to justify this approach P6 L2-10:

"The 24 hour deployment time was chosen to increase the likelihood to capture ebullition, which is episodic in nature, and to incorporate diel variability in diffusive emissions which can be up to a 2-fold bias (Bastviken et al., 2004;Bastviken et al., 2010;Natchimuthu et al., 2014). The use of long term deployments may underestimate diffusive fluxes, which decrease as the chamber headspace approaches equilibrium with the water. However, in contrast to $CO_2$, $CH_4$ has a long equilibration time and it has been shown that a 24 hour deployment of these types of flux chambers on lakes underestimate diffusive fluxes by less than 10% (Bastviken et al., 2010). An initial gas sample was collected at chamber deployment and a final chamber headspace gas sample after 24 hours following the Exetainer method described in Sturm et al., (2015)."

Measurements were all taken from the latter half of 2017 (Aug to Dec) and this can now be found P5 L20-21:

"…from August to December 2017…"

*Page 5, line 4: mention the number of ponds monitored, i.e. change to "The variability in surface area of each of the 22 ponds monitored in the emissions surveys was analyzed. . ."*

**Reply:** We have changed this section to include the sentence P6 L28-30:

"The variability in surface area of each of the 22 ponds monitored in the emissions surveys was analysed using high resolution historical imagery across all monitored water bodies."

*Page 6, line 16: is it possible to provide more quantitative evidence than "clearly" for the lognormal fit? In the Figure caption, a p-value is mentioned, but not in the text.*

**Reply:** We have removed "clearly" from the text and provided the p-values for the normal and log normal distributions in the manuscript text P8 L4-5:

"The emissions data from all replicate measurements fitted a log-normal (p = 0.081) but not a normal distribution (p = 0.0000)…"

*Page 6, line 28: what additional datasets?*

**Reply:** We have now referenced the Reservoirs and Water Storage Point datasets P8 L17-18:

"However, with the inclusion of the additional Reservoir and Water Storage Point datasets…"

*Page 6, line 35-36: Your first research objective was to quantify the area of ponds relative to regional assessments of larger artificial waterbodies – how does the 1,000 km2 compare to the total artificial water body surface area in Queensland?*

**Reply:** Thank you for highlighting this discrepancy. Larger artificial water bodies cover a surface area of 2,161 $km^2$, giving a total surface area of 3,248 $km^2$ and ponds make up 33% of the total surface area. This has now been included in the discussion P10 L12-14:

"Given ponds represent 33.5% of the total flooded lands surface area in Queensland and emission rates are equivalent to larger water bodies in the region (Musenze et al., 2014;Sturm et al., 2014), ponds represent one-third of total emissions from flooded lands in Queensland."

*Page 7, line 3: change mg m-2 d-1 to mg CH4 m-2 d-1 or mg CH4-C m-2 d-1 depending on which you mean. I'm guessing the former, but the latter is also sometimes used.*

**Reply:** This has now been corrected to mg $CH_4$ $m^{-2}$ $d^{-1}$ P8 L32.

*Page 8, line 13: remove "clearly"*

**Reply:** We have now removed "clearly" P10 L4.

*Page 8, line 14-15, 19-20: these results should be moved to section 3.1; however, the discussion of their importance is appropriate to have here*

**Reply:** We have moved these sentences to section 3.1 P8 L19-21:

"The official land use assessment of Queensland underestimates the surface area of ponds by 57%, and the total number of water bodies by more than an order of magnitude. The revised total surface area of all artificial water bodies across Queensland increased by 24% to just over 3,248 $km^2$ (Table A2)."

*Page 8, like 20-22: how do these emissions compare to mean annual CH4 emissions from larger inland waters in the state?*

**Reply:** Given emission rates from ponds are of a similar magnitude to those reported from larger water bodies, emissions from ponds are contributing one-third of total flooded land emissions in Queensland. This has now been included in the discussion P10 L12-14:

"Given ponds represent 33.5% of the total flooded lands surface area in Queensland and emission rates are equivalent to larger water bodies in the region (Musenze et al., 2014;Sturm et al., 2014), ponds represent one-third of total emissions from flooded lands in Queensland."

*Page 9, line 36: what do you mean by "available for emissions"? This phrasing is unclear. Consider changing to "as this will greatly improve the surface area estimate of flooded lands used for upscaling greenhouse gas emissions."*

**Reply:** Thank you for highlighting this and the suggested change has been made P11 L34-35:

[revised manuscript text omitted]

---

## Author Comment (AC3) · 25 Sep 2018

**General Comments:**

*This manuscript deals with an important issue, namely emissions of the strong greenhouse gas methane from the numerous small ponds in Queensland, Australia, in the context of climate changes and ponds number increase. Assessment of methane emissions supposes two steps: first the survey of small ponds and their relevant characteristics, second the assessment of methane emissions, depending on the pond characteristics (including size, type, location, climate and seasonality).*

**Reply:** Thank you for highlighting the importance of this issue, we believe artificial ponds are a major gap in the development of greenhouse gas inventories and aimed to highlight their potential as a greenhouse gas source. We appreciate the review comments, particularly those associated with the need for a histogram, and the revised manuscript has greatly improved. Below are our detailed responses to the review comments. Page and Line numbers refer to the attached revised manuscript which contains track changes for changes suggested by both reviewers.

*In this manuscript, the first step is realized based on existing data bases, the second on quite heavy emissions, and the dependence on emissions rate on inundated/non-inundated status of soil, the authors carried out complementary measures to characterize these causes of variability. The observations are then used to extrapolate emissions assessment to the global set of ponds, with two alternative extrapolation methods. This approach seems relevant. Anyway, several choices should have been discussed more in detail, and some underling hypothesis should have been clarified. In its present status, the manuscript is based on a large amount of data, not fully exploited. In particular, the way the "complementary" data is used (or not) is not clear.*

**Reply:** Thank you for these comments. The issue of global ponds emissions will need to be addressed once further regional studies have been completed. Our primary objective with this study was to establish the greenhouse gas status of a broad range of ponds in one (large) region, and then to attempt to quantify the overall magnitude of emissions within this region. The 'complementary data' was used to understand the relative importance of diffusive and ebullition pathways and provide a measure of variability of our scaled emission estimates. This study did not seek to explore the drivers of emissions, this will form the basis of our future research in this area. Drivers of ebullition are difficult to elucidate even with detailed datasets given the non-linear interactions that cause bubble release from sediments. The manuscript has now been revised and we have clarified our objectives further on P3 L19-28:

"The principle objective of this study was to establish the GHG status of ponds in Queensland, Australia. Given the paucity of GHG data from ponds, this study has focussed on empirical assessments of $CH_4$ emissions from a range of pond types rather than detailed assessments of drivers of these emissions. Our assessment comprised four components:

1. Quantify the area of ponds, relative to regional assessments of larger artificial water bodies;
2. Quantify $CH_4$ emission rates for a wide spectrum of pond types;
3. Determine variability in their surface area and emission rates;
4. Determine the influence of inundation level on emission rates.

When integrated together, these components provide a robust regional assessment of anthropogenic $CH_4$ emissions for ponds in Queensland, Australia."

We have identified the relevant areas of future research on P11 L8-13:

"An additional consideration for future studies of ebullition patterns in ponds stems from recent studies of reservoirs which found significant changes in ebullition intensity and ebullition distribution

as water levels decrease (Beaulieu et al., 2018;Hilgert et al., 2019). Under decreasing water levels, deeper zones of ponds may begin bubbling or increase the intensity of bubbling, this could potentially offset the reduction in surface available for emissions and total emissions would remain relatively constant."

And P12 L18-20:

"However, this finding was from a single urban lake and additional long term temporal studies along with high resolution spatial surveys of different pond types and size classes are required to identify the drivers of pond emission pathways."

**Specific comments:**

*Here are some more specific comments illustrating the above comments:*

***Introduction:***

*Introduction could have also cited the other kinds of greenhouse gases y emitted by artificial ponds (CO2, N2O), and could have evoked their potential role of organic carbon sink. Balancing these two antagonist influences would give a larger context to the rest of the manuscript.*

**Reply:** As suggested, we have now discussed $CO_2$ and $N_2O$ emissions in the introduction. There are no studies of greenhouse gas emissions from artificial ponds in the region, however larger artificial water bodies that have been studied have demonstrated the dominance of methane in GHG emissions, hence our focus on this greenhouse gas. This has been included in the introduction on P1 L33-35:

"Whilst carbon dioxide ($CO_2$), nitrous oxide ($N_2O$) and methane ($CH_4$) can all be emitted, the most recent global synthesis of artificial water body emissions demonstrated that when converted to $CO_2$ equivalents, $CH_4$ accounted for 80% of fluxes (Deemer et al., 2016)."

And P2 L36-38:

"…(Downing, 2010;Deemer et al., 2016) but are in agreement with assessments of larger water bodies where $CH_4$ is the dominant GHG relative to $N_2O$ and $CO_2$ (Merbach et al., 1996;Natchimuthu et al., 2014)."

The role of ponds as potential carbon sinks depends on the stability and permanence of organic carbon storage in sediments. In general they do not represent a stable long term sink, because they are routinely drained or the walls fail and the accumulated sediments are lost from the system. Furthermore, accumulations of organic sediment may enhance $CH_4$ emissions. These issues have now been discussed in more detail P2 L30-33:

"The potential of ponds as major organic carbon sinks has been established (Downing, 2010), although the stability and permanence of organic carbon trapped within ponds is critical to determining the magnitude of this sink. Loss pathways include active de-siltation (Verstraeten and Poesen, 2000), breaching of fully silted dams (Boardman and Foster, 2011) and methane emissions."

*P2 - l40, it would be interesting to give some orders of magnitude of the CH4 sink effect of soils prior inundation, to compare with the emission rates presented later.*

**Reply:** We have now included the order of magnitude rates for the methane sink in the manuscript, P3 L17:

"(ranging from -0.02 to -5 mg $CH_4$ $m^{-2}$ $d^{-1}$)"

*P3-l5: it seems a 5th point is missing, which is the extrapolation of the four other points results in a regional emission assessment: at this is not straightforward, it should be mentioned. Point 3 is a bit misleading, as spatial and temporal variability in emission rate is in fact assessed only for one pond, independently of the pond's area variation.*

**Reply:** We have removed the confusing terminology in component 3 and clarified that these four components are integrated into an overarching objective to quantify regional emissions from ponds in Queensland, Australia. This has section been revised on P3 L19-28:

"The principle objective of this study was to establish the GHG status of ponds in Queensland, Australia. Given the paucity of GHG data from ponds, this study has focussed on empirical assessments of $CH_4$ emissions from a range of pond types rather than detailed assessments of drivers of these emissions. Our assessment comprised four components:

1. Quantify the area of ponds, relative to regional assessments of larger artificial water bodies;
2. Quantify $CH_4$ emission rates for a wide spectrum of pond types;
3. Determine variability in their surface area and emission rates;
4. Determine the influence of inundation level on emission rates.

When integrated together, these components provide a robust regional assessment of anthropogenic $CH_4$ emissions for ponds in Queensland, Australia."

**2.1 Study area description**

*The link between the fact that the majority of artificial water bodies are less than 5 ML and the choice to study emissions from ponds which area is less than $10^5$ $m^2$ is not clear. Why this threshold?*

**Reply:** Previous studies of farm dams have shown that 90% of stock watering dams are less than 5 ML but we have now removed this statement and focussed on the use of the $10^5$ $m^2$ threshold. This has been revised on P4 L8-15:

"However, the number and surface area of ponds in Queensland is relatively unknown as there is no legal requirement to refer ponds to the state registry due to their small size.  Under current state law only dam walls in excess of 10 m and volumes above 750 ML are referable (DEWS, 2017) and the maximum reported volume for ponds in Queensland is three times less than the referable volume (< 250 ML) (SKM, 2012). This study has assumed ponds are less than 100,000 $m^2$ as this is recognised globally as the major area of uncertainty in surface area assessments (Lehner and Doll, 2004;Downing 2010) and has been identified as a threshold in global lake inventories (Downing et al., 2006;Verpoorter et al., 2014)."

**2.2 Relative surface area of ponds across the region**

*Given the discrepancy between the different sources of data and the difficulty to identify ponds and their characteristics within a large area, it would have been useful to give more details on the building of these three databases: how is data acquired, which kind of characteristic does each database include, at which periodicity is it revised, …*

**Reply:** This illustrates a major challenge in developing regional greenhouse gas inventories that incorporate ponds. The two State Government databases (Reservoirs and Water Storage Points)

were developed for Queensland to better understand the hydrology of the state in response to the Millennium Drought (1997 to 2009). They included a range of different aerial imagery (10 to 60 cm orthophotography) and satellite products (0.5 to 2.5 m resolution) in a major effort over the period of 2010 to 2014. Despite the availability of these high resolution databases these have still not been fully integrated into the official land use assessment for Queensland and represent a major area of uncertainty in the surface area available for emissions. This is the first time such an exercise has been attempted so the periodicity at which it will be revised is difficult to estimate. However, with high resolution satellite imagery becoming increasingly available this will allow the total surface area of ponds to be revised more frequently.

We have added additional information on these databases and how they were used in P4 L21-23:

"To determine the number and relative surface area of ponds across the Queensland, three State Government GIS databases of artificial water bodies were utilised. However these databases required additional processing to extract comparable pond data as there were inconsistencies in the format and nomenclature of feature types."

And P4 L31-32:

"Both databases are derived from aerial (10 to 60 cm orthophotography) and satellite (0.5 to 2.5 m resolution) imagery captured between 2010 and 2014."

*Given the strong dependency of CH4 emission rate on the ponds' size, why choosing an average area for ponds <625m², instead of using a size distribution. The same question arises for the classification in the three classes of the GRanD. This seems premature before the later results. Anyhow, it would have been interesting to present a histogram of the ponds' sizes. It is also disappointing not to know more about the types of ponds (and the potential link between type and size), their location, the way they are supplied in water, ... all characteristic which may influence methane emissions and which may be available in the databases?*

**Reply:** We share the frustration at not having more information about the type of ponds, but for both databases there was very limited classification of feature types: Reservoirs were divided into feature types: Flood Irrigation Storage, Rural Water Storage and Town Water Storage. Water Storage Points only had one feature type for all points: Dam. The ponds in Water Storage Points database were classified in the $10^2$ to $10^3$ $m^2$ size class and there was a very weak relationship between median emission rate and pond surface area (see figure below). For this reason we preferred to use a mean surface area for these systems.

[Figure]

We have also included a histogram showing the relationship between the three major pond types and their sizes in the Appendix P30 Figure A2:

[Figure]

**Figure A2. Pond size from emission study relative to histogram of regional pond distribution of stock dams, irrigation dams and urban lakes. The surface area of pond used emission study (Table A1). Histogram of regional distribution of ponds was developed from QLUMP, Reservoir and Water Storage Points databases and separated into pond type depending on surrounding land use: "Grazing native vegetation" for stock dams; "Production from irrigated agriculture and plantations" for irrigation dams; "Intensive uses" for urban lakes with "Mining" and "Manufacturing" landuse within "Intensive Uses" were removed to ensure only urban areas were selected. To incorporate the distribution of ponds within the Water Storage Points database, it**

**was assumed this would match the distribution from the 100 individual ponds examined in Section 2.2 to determine their average surface area.**

**2.3 CH4 emissions from broad spectrum of pond types**

*It would have been useful for readers not familiar with methane emission from waterbodies to present the different kind of methane emissions measures, their advantages and limits, and to argument the choice performed here. Above all, the choice of the studied ponds should have been discussed, and their representativeness of the whole ponds set variety assessed.*

**Reply:** We have now included this suggestion and revised the methodology to include the relative advantages and disadvantages of commonly used methods to assess methane emissions and to justify the approach we took in our choice to use floating chambers.  These changes are on P5 L23-25:

"Stock dams, irrigation dams and urban lakes account for the vast majority of ponds across Queensland and ponds within each category were selected to represent the regional size class distribution (Fig. A2)."

And P5 L27-33:

"There are a number of commonly used methods to assess methane emissions from water bodies depending on the pathway of interest. For the diffusive emission pathway, rates may be modelled using the thin boundary methods or directly measured using manual or automatic floating chambers (St. Louis et al., 2000). For ebullition pathways, rates can be directly measured using acoustic surveys or funnel traps (DelSontro et al., 2011). Thin boundary layer models cannot be used to quantify the ebullition pathway and acoustic surveys or funnel traps cannot be used effectively in ponds as the water depth is often too shallow (< 1 m). We chose to use floating chambers to capture both ebullition and diffusive fluxes."

Small, light chambers had an additional advantage in their relative ease of deployment in these relatively challenging environments, examples of which are now provided in the Appendix P31 Figure A4:

[Figure]

**Figure A4. Oblique drone images showing natural obstacles for pond chamber deployments from a) emergent macrophytes and b) floating aquatic weeds.**

The representativeness of the studied ponds relative to the whole region is now shown on the histogram in the Appendix P30 FigureA2, where the studied ponds captured the regional surface area distribution for major pond types.

*How was the number of floating chambers per pond chosen? It seems not to be only in function on the pond's size? Was the uncertainty arising from measuring only 6 to 8 hours of emission for 3 ponds assessed? Is the emission process known to be varying at the daily scale? When did the monitoring occurred during the year (and which year rather dry or wet)?*

**Reply:** We have now included more detail in the methodology to address these comments. The number of chambers deployed per pond was a function of both the pond size and local restrictions on access. For each pond a minimum of 3 chambers were deployed along a transect to ensure deep and shallow areas were monitored. The uncertainty arising from 6 to 8 hours deployment was not assessed, however, studies to date have shown diffusive emissions may be subject to diurnal changes but ebullition (typically the largest emission pathway) does not undergo regular diurnal change, so there should not be a significant bias. Of the three ponds where 6 to 8 hour incubation were undertaken, the diffusive pathway was only dominant at Lake Alford. More detail on the uncertainty has been stated in P6 L2-8:

"The 24 hour deployment time was chosen to increase the likelihood to capture ebullition, which is episodic in nature, and to incorporate diel variability in diffusive emissions which can be up to a 2-fold bias (Bastviken et al., 2004;Bastviken et al., 2010;Natchimuthu et al., 2014). The use of long term deployments may underestimate diffusive fluxes, which decrease as the chamber headspace approaches equilibrium with the water. However, in contrast to $CO_2$, $CH_4$ has a long equilibration

time and it has been shown that a 24 hour deployment of these types of flux chambers on lakes underestimate diffusive fluxes by less than 10% (Bastviken et al., 2010)."

The annual study was undertaken in 2017 and the broad water body survey measurements were undertaken in August to December 2017.  This has been stated P5 L20-21 ("from August to December 2017"). Rainfall across Queensland in 2017 was 10% below average.

*Nothing is said about the way the pond global emission rate is assessed from the punctual measures: given the variability at the pond scale illustrated with 2.4.1 results, it yet seems crucial. Uncertainty arising from how this calculation was performed may be as high as those arising from the choice of arithmetical or geometrical means between ponds' emission rates later on (2.6).*

**Reply:** We took the approach to use the average surface area at full supply level ($A_{FSL}$) for respective size classes. We believe the highest uncertainty lies with the emissions rates as these varied by over an order of magnitude whereas as the surface area variability, particularly in the larger pond size classes was relatively stable, and the mean ranged only 5%. The uncertainty associated with surface area changes is buffered by the consistency of emissions from deeper zones which experience less frequent drying periods. These comments are now included in the discussion sections P11 L8-13:

"An additional consideration for future studies of ebullition patterns in ponds stems from recent studies of reservoirs which found significant changes in ebullition intensity and ebullition distribution as water levels decrease (Beaulieu et al., 2018;Hilgert et al., 2019). Under decreasing water levels, deeper zones of ponds may begin bubbling or increase the intensity of bubbling, this could potentially offset the reduction in surface available for emissions and total emissions would remain relatively constant."

**2.4 Spatial and temporal variability in surface area and emission rate**

**2.4.1**

*Here also it is not clear why this pond was chosen rather than another. What about its representativeness? For example, one can expect that the temporal variability of a weir's emission rate to be higher than an urban's lake one? The year when this monitoring was performed is not specified, neither corresponding rainfall, which may be of influence on the pond supply and the emission processes? Air and water temperature may also be influent factors?*

**Reply:** This pond was chosen for both its ease of access as well as relatively stable surface area allowing long term monitoring of the same site. This has been clarified on P6 L17-19:

"This pond was selected as water level remains relatively constant throughout the year and sampling would not impacted by changes in inundation status."

It provides a typical example of an urban lake but we recognise and acknowledge that we would need measurements from examples of each pond type to provide a truly representative dataset. Rainfall at the urban lake site was average for 2017 and there were no major maintenance programs undertaken on the lake during this time. Our aim was to obtain indicative and upscaleable estimates of emissions in this study, not to fully elucidate the mechanisms determining temporal variability which would require a different approach. This will form the basis of future studies for our research in this area. We have now adjusted the future research section to include the need for focussed studies to establish the major drivers of emissions within each pond type P12 L18-20:

"However, this finding was from a single urban lake and additional long term temporal studies along with high resolution spatial surveys of different pond types and size classes are required to identify the drivers of pond emission pathways."

**2.4.2**

*Is rainfall variability homogeneous at the state scale? In other words, are the percentages of AFSL calculated for the ponds which were monitored relevant for the regional scaling which is performed in 4.1?*

**Reply:** Rainfall is not homogenous at the state scale, with clear zones of variability following the isohyets shown in Figure 2 b. Low rainfall areas (< 300 mm yr$^{-1}$) particularly in the western parts have the highest variability, whilst zones (> 300 mm yr$^{-1}$) closer to the coastal regions have low to moderate variability (http://www.bom.gov.au/jsp/ncc/climate_averages/rainfall-variability/index.jsp). Over 90% of the state's ponds are located within the bands of low to moderate rainfall variability and would support regional scaling from the ponds measured in this study. To calculate the percentage of $A_{FSL}$ imagery was collected spanning the time period 2009 to 2017 which includes the both drought and flood years (P6 L35-37) and would cover the expected range in rainfall across the state and provides additional confidence in the regional scaling.

**2.5**

*As for the 2.4.1 section, it is not clear why this pond was chosen rather than another. What assures its representativeness of other small ponds which area varies a lot depending on rainfall?*

**Reply:** This pond was selected to represent stock dams which are the most numerous pond type and represent the largest surface area (Appendix P30 Figure A2). The construction of thi pond is typical of those in the region (a shallow pit is dug out and the soil used to construct the wall and spillway) and the surface area closely matched the median for all farm dams. The variability in surface area for stock ponds in this region is primarily dependent on whether they are being used stock watering, rather than rainfall. The site justification is now provided on P7 L5-9:

"This pond was selected as stock dams generally experience accelerated rates of water level change due to their relatively small size compared to other pond types (Fig. A2). In addition, the construction of this pond is typical for stock dams (a shallow pit is dug out and the soil used to construct the wall and spillway) and the surface area (1,893 m$^2$) closely matched the median for all farm dams (1,586 m$^2$; Fig. A2)."

*These complementary field campaigns are honest and interesting attempts to deepen the study, but should be more detailed and argued to be totally useful and convincing to strengthen the results which are the main scope of the paper.*

**Reply:** We are very grateful for your insights and have clarified the introduction and study objectives to support our approach. This can be found on P2 L8-11:

"In addition, the peripheral areas of small water bodies regularly experience periods of inundation and no inundation as water levels change due to their relatively shallow nature and high water use rates. The changes in their inundation status may influence emission rates as has been observed for natural ponds (Boon et al 1997)."

And P3 L4-6:

"An important part of the value of building a dataset of $CH_4$ flux estimates from a broad range of sites is determining factors that account for spatial and temporal variability in the flux."

And P3 L19-21:

"The principle objective of this study was to establish the GHG status of ponds in Queensland, Australia. Given the paucity of GHG data from ponds, this study has focussed on empirical assessments of $CH_4$ emissions from a range of pond types rather than detailed assessments of drivers of these emissions."

Furthermore, we revised the methods, results and future studies to demonstrate the utility of these complementary studies. This is stated in the following sections on:

P6 L14-16:

"To gain insight into the spatial and temporal uncertainty in pond emissions we compared variability in seasonal emissions from a single site to emissions from an intensive spatial survey of multiple sites across the pond (Fig. 4)."

P9 L31-33:

"In contrast emissions from central areas were over 100 mg m$^{-2}$ d$^{-1}$, more than double the peripheral area emission rates (Table A1)."

P12 L14-15:

"However, this was limited to a single stock dam and additional pond types and size classes must be examined before more confident generalisations can be made."

P12 L18-20:

"However, this finding was from a single urban lake and additional long term temporal studies along with high resolution spatial surveys of different pond types and size classes are required to identify the drivers of pond emission pathways."

**3.2 CH4 emissions from ponds**

*Some considerations on ebullition/diffusive emissions would be necessary to support the conclusion that ebullition is the dominant emission pathway. If so, spatial heterogeneity of emission must be high: is the monitoring protocol adapted to capture this heterogeneity?*

**Reply:** We agree that it is important to capture both the spatial and temporal heterogeneity in the ebullition pathway. We believe that our approach to follow a transect across deep and shallow areas should capture this spatial variability at a broad level, and that the 24 hour deployments would maximise the likelihood of capturing episodic ebullition events. We accept that both the total $CH_4$ emissions and the balance of diffusion and ebullition are spatially variable and uncertain, and have acknowledged this in the manuscript P5 L27-33:

"There are a number of commonly used methods to assess methane emissions from water bodies depending on the pathway of interest. For the diffusive emission pathway, rates may be modelled using the thin boundary methods or directly measured using manual or automatic floating chambers (St. Louis et al., 2000). For ebullition pathways, rates can be directly measured using acoustic surveys or funnel traps (DelSontro et al., 2011). Thin boundary layer models cannot be used to quantify the ebullition pathway and acoustic surveys or funnel traps cannot be used effectively in ponds as the

water depth is often too shallow (< 1 m). We chose to use floating chambers to capture both ebullition and diffusive fluxes."

And P5 L37-39:

"The floating chambers used were designed to yield negligible bias on the gas exchange and compare well with non-invasive approaches (Cole et al., 2010;Gålfalk et al., 2013;Lorke et al., 2015)."

And P6 L2-8:

"The 24 hour deployment time was chosen to increase the likelihood to capture ebullition, which is episodic in nature, and to incorporate diel variability in diffusive emissions which can be up to a 2-fold bias (Bastviken et al., 2004;Bastviken et al., 2010;Natchimuthu et al., 2014). The use of long term deployments may underestimate diffusive fluxes, which decrease as the chamber headspace approaches equilibrium with the water. However, in contrast to $CO_2$, $CH_4$ has a long equilibration time and it has been shown that a 24 hour deployment of these types of flux chambers on lakes underestimate diffusive fluxes by less than 10% (Bastviken et al., 2010)."

On the other hand, existing data on $CH_4$ emissions from artificial ponds are extremely scarce, to the extent that this study alone will considerably increase the total global dataset of measurements. We hope that it will therefore represent an important step towards developing a more comprehensive understanding of the role ponds play in greenhouse gas emissions and the carbon cycle. This has been included in sections introduction and future areas of research:

P2 L29-33:

"In addition, quantifying methane emission from ponds will improve our understanding of their role in the global carbon cycle. The potential of ponds as major organic carbon sinks has been established (Downing, 2010), although the stability and permanence of organic carbon trapped within ponds is critical to determining the magnitude of this sink. Loss pathways include active de-siltation (Verstraeten and Poesen, 2000), breaching of fully silted dams (Boardman and Foster, 2011) and methane emissions."

And P11 L34-35:

"… will greatly improve the surface area estimate of flooded lands used for upscaling greenhouse gas emissions as well as their role in the global carbon cycle."

*As said before, a histogram of sizes and types of ponds would be useful to contextualize the results. Detail of the way the emissions are assessed at the pond scale too.*

**Reply:** This is an excellent suggestion, thank you - we have now added the suggested histogram, which indicates that our sample sites closely reflect the mix of pond types and areas within our study region Appendix P30 Figure A2.

*As weirs present high emission rates compared to other types of ponds, is it relevant to group them with other small ponds/stock ponds to assess the regional scale emissions?*

**Reply:** We accept this comment, and have now excluded weirs from the regional emissions scaling and revised table accordingly (P18 Table 1). The regional contribution decreased by less than 1% and the study conclusions remain valid. Revised text is found on P18 L8 "however, weir emissions were omitted as these are not relevant at the regional scale."

**3.3.1 Spatial and temporal variability within a single pond**

*These data are no doubt very interesting, but what they bring to this study is not clear for me. How are they used to the following regional scaling? If it is by considering that observed emissions from the other ponds can be taken as annual averages, this should be clarified. If this is the case, the validity of this hypothesis should be discussed, as only one type of pond was considered for this analysis. It seems to me that this part could be cut from this manuscript, and maybe give the material for another paper, which would allow to analyse more in depth the emissions variability and the factor which influence it.*

**Reply:** We have now clarified the text to highlight the limitations of extrapolating these results across different pond types. The value we see in these data is demonstrating ebullition across the annual cycle at a single point, suggesting this is a persistent feature when conditions favour it. Additional text can be found on P12 L18-20:

"However, this finding was from a single urban lake and additional long term temporal studies along with high resolution spatial surveys of different pond types and size classes are required to identify the drivers of pond emission pathways."

The spatial and temporal patterns observed in these data do provide striking similarities with larger reservoirs in the region supporting the persistence of the ebullition pathway across the annual cycle and the importance of spatial variability in emissions rates. Both of which we consider valuable insights into the emissions from a smaller water body. On balance we would therefore prefer to retain this part of the manuscript.

**4.1**

*In India, numerous ponds are used to increase groundwater discharges. As a consequence, CH4 emission from these ponds may differ from the types of ponds which were studied here. This could be specified to be rigorous in this discussion about the importance to take into account ponds in methane emission rates assessment.*

**Reply:** Thank you for this suggestion and we have now included a section to highlight the importance of local pond types to regional emissions P12 L4-7:

"An additional consideration is to ensure pond emission studies from different regions include all relevant ponds types. For example, the use of ponds to increase groundwater recharge is widespread across South East Asia (Giordano, 2009) and these would need to be included in regional inventories."

**4.2**

*Again, more details should have been given above on the different emission pathways and their influencing factors.*

**Reply:** We have now included sections providing more details on the major pathways and their influencing factors as well as key areas for future research to address these issues. These sections can be found on P11 L8-13:

"An additional consideration for future studies of ebullition patterns in ponds stems from recent studies of reservoirs which found significant changes in ebullition intensity and ebullition distribution as water levels decrease (Beaulieu et al., 2018;Hilgert et al., 2019). Under decreasing water levels, deeper zones of ponds may begin bubbling or increase the intensity of bubbling, this could potentially offset the reduction in surface available for emissions and total emissions would remain relatively constant."

And P12 L4-7:

"An additional consideration is to ensure pond emission studies from different regions include all relevant ponds types. For example, the use of ponds to increase groundwater recharge is widespread across South East Asia (Giordano, 2009) and these would need to be included in regional inventories."

And P12 L18-20:

"However, this finding was from a single urban lake and additional long term temporal studies along with high resolution spatial surveys of different pond types and size classes are required to identify the drivers of pond emission pathways."

*As depth, the way water is supplied in pond and substrate seem to be influential, this discussion could address the question of the availability of such data in current databases.*

**Reply:** Thank you for this comment and, unfortunately, no depth data or the water delivery information is provided in the databases. Depth can be inferred from surface area to volume relationships such as that reported in Lowe et al 2005. Water supply occurs through a number of different pathways including overland flow from rainfall, direct pumping from river and stormwater inflow from artificial drainage networks.

Lowe, L., Nathan, R., and Morden, R. 2005. Assessing the impact of farm dams on streamflows, Part II: Regional characterisation, Australasian Journal of Water Resources, 9, 13-26.

**5 Future research**

*Some sources of uncertainties are taken into account in this manuscript, other are not: this section could emphasize these points, and discuss how to handle them (assessment of a whole pond emission rate given punctual data in time and space; identification of emission pathways, characterisation of the way some factors influence emission rates–type, depth, purpose, water supply, temperature …) . At the moment, the research perspectives are more or less an extension of the work which was already performed.*

**Reply:** We have now included a broader discussion of the sources of uncertainty, as suggested, as well as the priority research needs required to reduce this uncertainty and improve our understanding of pond's contribution to flooded land emissions and loss pathways of sequestered carbon. New text has been added in the following sections:

P2 L29-33:

"In addition, quantifying methane emission from ponds will improve our understanding of their role in the global carbon cycle. The potential of ponds as major organic carbon sinks has been established (Downing, 2010), although the stability and permanence of organic carbon trapped within ponds is critical to determining the magnitude of this sink. Loss pathways include active de-siltation (Verstraeten and Poesen, 2000), breaching of fully silted dams (Boardman and Foster, 2011) and methane emissions."

P11 L8-13:

"An additional consideration for future studies of ebullition patterns in ponds stems from recent studies of reservoirs which found significant changes in ebullition intensity and ebullition distribution as water levels decrease (Beaulieu et al., 2018;Hilgert et al., 2019). Under decreasing water levels, deeper zones of ponds may begin bubbling or increase the intensity of bubbling, this could

potentially offset the reduction in surface available for emissions and total emissions would remain relatively constant."

And P12 L4-7:

"An additional consideration is to ensure pond emission studies from different regions include all relevant ponds types. For example, the use of ponds to increase groundwater recharge is widespread across South East Asia (Giordano, 2009) and these would need to be included in regional inventories."

And P12 L14-15:

"However, this was limited to a single stock dam and additional pond types and size classes must be examined before more confident generalisations can be made."

And P12 L18-20:

"However, this finding was from a single urban lake and additional long term temporal studies along with high resolution spatial surveys of different pond types and size classes are required to identify the drivers of pond emission pathways."

Incorporating the comments from both reviews has improved this manuscript and highlighted the importance of this work in highlighting the role ponds likely play in global greenhouse gas emissions.

[revised manuscript text omitted]